# Archaeal DNA-import apparatus is homologous to bacterial conjugation machinery

Leticia C. Beltran[1], Virginija Cvirkaite-Krupovic[2], Jessalyn Miller[3], Fengbin Wang[1,7], Mark A. B. Kreutzberger[1], Jonasz B. Patkowski[4], Tiago R. D. Costa[4], Stefan Schouten[5], Ilya Levental[6], Vincent P. Conticello[3], Edward H. Egelman[1]✉ & Mart Krupovic[2]✉

Conjugation is a major mechanism of horizontal gene transfer promoting the spread of antibiotic resistance among human pathogens. It involves establishing a junction between a donor and a recipient cell via an extracellular appendage known as the mating pilus. In bacteria, the conjugation machinery is encoded by plasmids or transposons and typically mediates the transfer of cognate mobile genetic elements. Much less is known about conjugation in archaea. Here, we determine atomic structures by cryo-electron microscopy of three conjugative pili, two from hyperthermophilic archaea (*Aeropyrum pernix* and *Pyrobaculum calidifontis*) and one encoded by the Ti plasmid of the bacterium *Agrobacterium tumefaciens*, and show that the archaeal pili are homologous to bacterial mating pili. However, the archaeal conjugation machinery, known as Ced, has been 'domesticated', that is, the genes for the conjugation machinery are encoded on the chromosome rather than on mobile genetic elements, and mediates the transfer of cellular DNA.

The importance of horizontal gene transfer (HGT) in microbial persistence and evolution cannot be overstated. Exchange of genetic information is essential for the survival of microbial populations that otherwise succumb to Muller's ratchet, a process whereby the irreversible accumulation of deleterious mutations leads to extinction of an asexual population[1,2]. Furthermore, HGT plays a major role during the adaptation of microbes to constantly changing environmental conditions by providing immediate access to beneficial traits and promoting cooperation within microbial communities[3]. Accordingly, bacteria and archaea have evolved dedicated mechanisms of HGT[4,5]. Traditionally, three major routes of HGT are recognized, namely, natural transformation, transduction, and conjugation. Whereas transformation is a natural ability of cells to uptake exogenous DNA from the environment through a dedicated competence system[6–8], the other two HGT mechanisms rely on distinct types of mobile genetic elements (MGE), viruses and plasmids (or integrative and conjugative elements), respectively. An additional HGT route, which is gaining increasing recognition, is intercellular DNA transfer through membrane-bound extracellular vesicles[9–12].

In bacteria, conjugation is one of the main mechanisms for the spread of antibiotic resistance and other adaptive traits[13,14]. Conjugation requires a sophisticated MGE-encoded apparatus, which belongs to the type IV secretion system (T4SS) superfamily, and in diderm bacteria consists of four key components: (i) a conjugative pilus, a multimeric assembly of the major pilin protein, which connects the donor and recipient cells and serves as a conduit for DNA transfer; (ii) the type IV coupling protein, an AAA + ATPase essential for pilus biogenesis and substrate transfer; (iii) the T4SS membrane-spanning

[1]Department of Biochemistry and Molecular Genetics, University of Virginia, Charlottesville, VA 22903, USA. [2]Institut Pasteur, Université Paris Cité, CNRS UMR6047, Archaeal Virology Unit, 75015 Paris, France. [3]Department of Chemistry, Emory University, Atlanta, GA 30322, USA. [4]MRC Centre for Molecular Bacteriology and Infection, Department of Life Sciences, Imperial College, London, UK. [5]NIOZ Royal Netherlands Institute for Sea Research, Department of Marine Microbiology and Biogeochemistry, Texel, The Netherlands. [6]Department of Molecular Physiology and Biological Physics, Center for Membrane and Cell Physiology, University of Virginia, Charlottesville, VA 22903, USA. [7]Present address: Department of Biochemistry and Molecular Genetics, University of Alabama Birmingham, Birmingham, AL 35233, USA. ✉e-mail: egelman@virginia.edu; mart.krupovic@pasteur.fr

protein complex enabling DNA transfer across the membrane of the donor cell; and (iv) the relaxosome, which nicks the double-stranded DNA (dsDNA), yielding the single-stranded DNA (ssDNA) substrate for intercellular transfer[15–19]. Conjugative elements have been identified as extrachromosomal plasmids or as integrated elements in certain archaea, including hyperthermophilic archaea of the order Sulfolobales[20–24] and ammonia-oxidizing archaea of the class Nitrososphaeria[25], but the mechanism of conjugation has not been investigated in detail. Notably, none of the archaeal conjugative plasmids encode recognizable homologs of the relaxase or pilus protein and it has been suggested that the mechanism of conjugation in archaea might be different from that operating in bacteria[23].

Hyperthermophilic archaea of the order Sulfolobales have evolved a distinct DNA transfer system, named crenarchaeal exchange of DNA (Ced), which is dependent on species-specific cell aggregation and is inducible upon UV irradiation[26]. The Ced system operates in conjunction with the UV-inducible type IV pili operon of Sulfolobales (Ups) system[27]. The Ups pili produced upon UV irradiation mediate cellular aggregation in a species-specific manner, ensured by specific glycosylation patterns on the Ups pili and the protein S-layer, which covers the cellular membrane[28]. Both Ced and Ups systems are required for efficient DNA exchange, but the two do not have to be expressed in the same cell[26]. Notably, the Ced system mediates unidirectional import of DNA, which is then used as a template for genome repair by homologous recombination; cells that cannot exchange DNA show significantly lower survival rates upon DNA damage[29]. Notably, some crenarchaeal species encode the Ced but not the Ups system, whereas others, such as members of the order Thermoproteales, have not been found to encode either[26], suggesting alternative mechanisms for DNA exchange.

The Ced system consists of four proteins, CedA, CedA1, CedA2, and CedB[26]. CedA contains six or seven transmembrane domains and is believed to form a transmembrane channel for DNA import, whereas CedB is homologous to VirB4/HerA-like AAA + ATPases and appears to

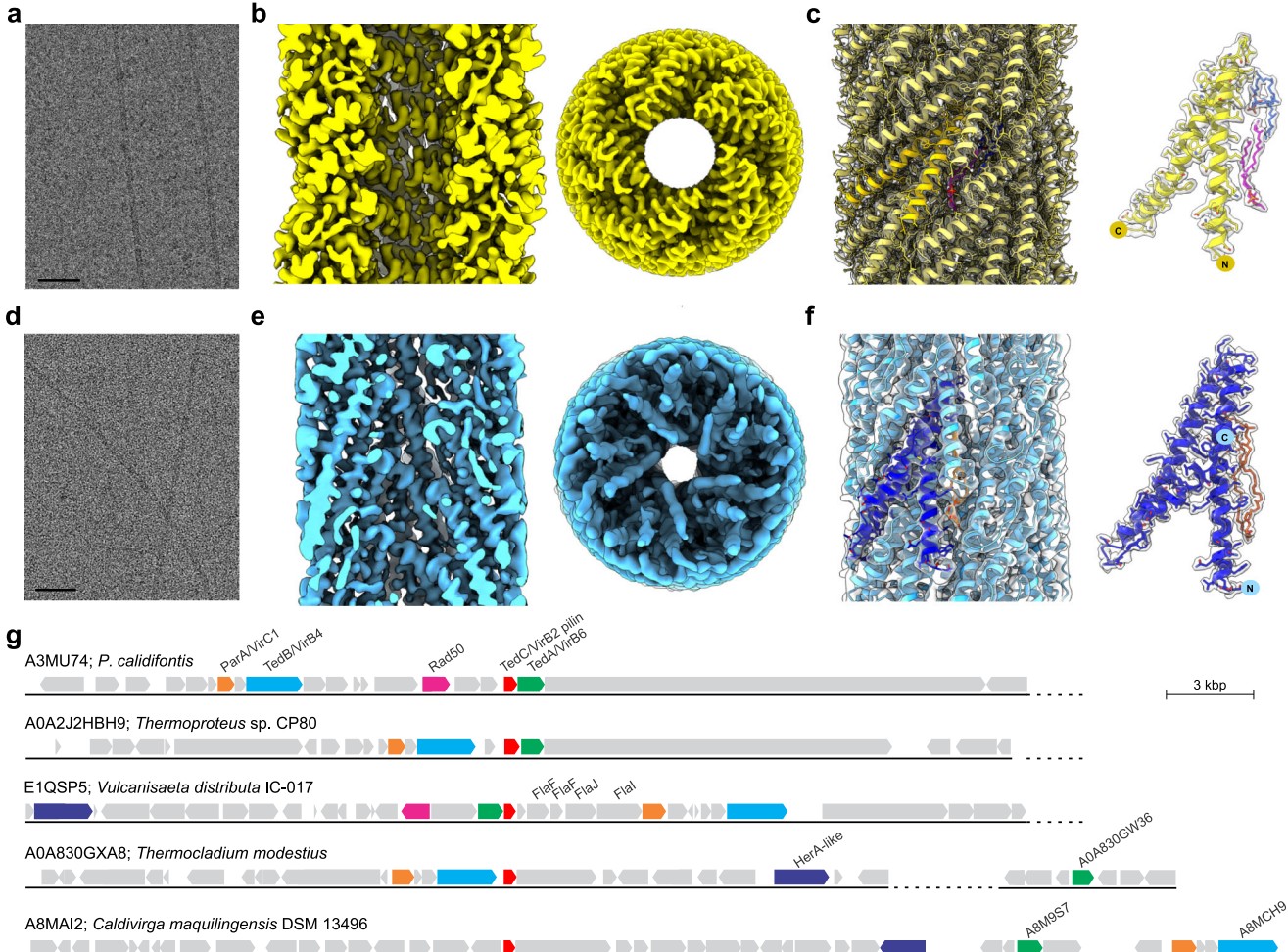

**Fig. 1 | Archaeal conjugative pili. a, d** Representative cryo-electron micrographs of *A. pernix* and *P. calidifontis* pili, respectively, from >8000 images collected for each. Scale bars, 50 nm. **b, e** Side and top views of the *A. pernix* (**b**, yellow) and *P. calidifontis* (**e**, blue) cryo-EM density maps at resolutions of 3.3 Å and 4.0 Å, respectively. The front half of the filament has been removed in the side views, so that lumens are visible. **c, f** Atomic models in ribbon representation of the *A. pernix* (**c**, yellow) and *P. calidifontis* (**f**, blue) pili docked within their respective transparent cryo-EM density maps. The asymmetric unit of the *A. pernix* pilin CedA1 (**c**, bright yellow) shows two bound lipids (magenta and blue), while a single asymmetric unit of the *P. calidifontis* pilin TedC (**f**, dark blue) shows one bound tetraether lipid (orange). **g** Genomic loci encompassing the Ted system in different members of the order Thermoproteales. Each of the five genera within the Thermoproteales (genera *Caldivirga, Pyrobaculum, Thermocladium, Thermoproteus,* and *Vulcanisaeta*) is represented. Genes encoding the conserved VirB2/CedA1-like pilin protein TedC, VirB6/CedA-like transmembrane channel TedA, and VirB4/CedB-like AAA + ATPase TedB are shown as red, green, and cyan arrows, respectively. Additional conserved genes are also indicated, including VirC1/ParA-like, Rad50, and HerA-like helicase which are shown as orange, magenta, and dark blue arrows, respectively. Other genes are shown in gray. Genomic loci are aligned using the TedC gene and indicated with the corresponding UniProt accession numbers, followed by the organism name. In some species, the components of the Ted systems are encoded within distal genomic loci which are separated from the TedC-encoding loci by dashed lines, and the corresponding genes are identified with their UniProt accession numbers.

power the DNA translocation across the membrane. The function of CedA1 and CedA2, each with two predicted transmembrane domains, is less clear, but they were shown to form a membrane-localized complex with CedA[26]. The Ced system was considered to be unrelated to the bacterial conjugation system because of the opposite directionality of DNA transfer (import versus export, respectively) and the lack of homologs other than VirB4-like ATPase[26]. However, how the DNA is transported between the cells within the Ups pili-mediated cellular aggregates and the nature of the channel connecting the donor and recipient cells remained unresolved. It also remained unclear whether the Ced system transfers dsDNA or ssDNA substrates.

Here, using cryo-electron microscopy (cryo-EM), we show that a protein from the hyperthermophilic archaeon *Aeropyrum pernix*, a homolog of CedA1, forms a pilus which is structurally homologous to bacterial conjugative pili. We also discover that structurally similar pili, although with no sequence similarity, are produced by members of the Thermoproteales, which were previously not considered to encode the Ced-like system. We present high-resolution structures of two putative conjugative pili from hyperthermophilic archaea, *A. pernix* and *Pyrobaculum calidifontis*, and a bacterial conjugation pilus from a model system[30–33], encoded by the pTiC58 plasmid of *Agrobacterium tumefaciens*[34]. It has been previously stated that the pilin subunit in the *A. tumefaciens* mating pilus is cyclic[35,36], and that this accounts for its robust stability[37]. We show that it is not cyclic and is actually similar in fold to other bacterial and archaeal mating pili. Collectively, our results suggest that the archaeal Ced-like systems share a common ancestor with bacterial T4SS conjugation system. However, unlike in bacteria, where conjugation systems are proprietary to mobile genetic elements, in hyperthermophilic archaea the DNA transfer system has been domesticated, and we propose that this has evolved to ensure survival in extreme environments.

## Results

### Identification of putative DNA transfer pili in hyperthermophilic archaea

In Sulfolobales, expression of the *ced* and *ups* genes is activated exclusively upon UV irradiation[26]. We set out to study the behavior of the Ced system in *Aeropyrum pernix* (order Desulfurococcales), a hyperthermophilic archaeon that grows at temperatures up to 100 °C[38] and lacks the Ups system[26]. Given that intercellular DNA transfer typically involves extracellular filaments, the extracellular fraction of *A. pernix* cells was analyzed using cryo-EM. In addition to the flagella[39], we identified a new type of filament (Fig. 1a), not previously observed in archaea. The reconstruction of this pilus to 3.3 Å resolution allowed us to determine the pilin identity directly from the cryo-EM map. The pilin was identified as *A. pernix* protein APE_0220a (WP_010865579), an ortholog of the *S. acidocaldarius* protein CedA1 (WP_011277463), one of the conserved components of the Ced system previously thought to be an integral membrane protein[26].

We have previously shown that *Pyrobaculum calidifontis*, a hyperthermophilic archaeon of the order Thermoproteales[40], which lacks both Ced and Ups systems, is prone to aggregation mediated by bundling pili related to TasA-like fibers, a major component of the biofilm matrix in many bacteria[41]. We thus explored whether *P. calidifontis* cells produce pili which could be involved in DNA transfer. Cryo-EM analysis of the *P. calidifontis* filament preparation revealed pili (Fig. 1d), which following the reconstruction to 4.0 Å resolution (Fig. 1e), proved to be structurally similar to the CedA1 pili of *A. pernix* (Fig. 1b, Supplementary Fig. 1a). While in the *A. pernix* filament the helical rise and twist per subunit were 3.6 Å and 76.5°, respectively, in the *P. calidifontis* filament these parameters were 5.0 Å and 74.2°. From the secondary structure and side-chain information present, we were able to determine the pilin identity directly from the cryo-EM map using DeepTracer-ID[39] to be *P. calidifontis* protein Pcal_0765 (WP_011849449) (Fig. 1f), which we name TedC (see below). Although

TedC displays a similar fold to CedA1 from *A. pernix* (Supplementary Fig. 1a), the two pilins are processed differently, with *A. pernix* pilin not being processed and the *P. calidifontis* pilin, similar to bacterial plasmid conjugative pilins, undergoing proteolytic cleavage. Indeed, SignalP analysis[42] predicts that TedC carries a cleavable signal peptide, with the predicted signal peptidase I cleavage site, 20- AVA ↓ QA-24 (cleavage site probability of 0.97; Supplementary Fig. 1b). Since the atomic model built into the reconstruction starts at residue 38, rather than the predicted residue 23, we conclude that residues 23-37 are disordered and therefore not visualized in the density map.

Sequences homologous to TedC were identified using BLASTP searches in members of all five genera of the order Thermoproteales, namely, *Pyrobaculum*, *Thermoproteus*, *Caldivirga*, *Vulcanisaeta,* and *Thermocladium*. Genomic neighborhood analysis (Fig. 1g) showed that the gene downstream of the *tedC* encodes a protein with seven predicted transmembrane domains, similar to CedA of *S. acidocaldarius* and *A. pernix*. Although BLASTP searches did not reveal the relationship between Pcal_0766 and CedA, sensitive profile-profile comparisons showed that the two proteins are indeed homologous (HHpred probability: 98.6), despite negligible pairwise sequence identity of 13% (Supplementary Fig. 2a). Notably, profile-profile comparisons of Pcal_0766 against the PDB database showed that it is distantly related to VirB6-like proteins encoded by bacterial conjugative plasmids and involved in formation of the mating pore complex (Supplementary Fig. 2b). A gene encoding the VirB4-like ATPase was identified transcriptionally upstream of the *tedC* and *cedA*-like genes, separated by a few genes (Fig. 1g). In *Thermocladium* species, the ortholog of *Pyrobaculum virB4*-like gene is adjacent to the *tedC* pilin gene, suggesting that the corresponding proteins function together. Given the high sequence divergence between the components of the Ced system of Sulfolobales and the related system of Thermoproteales, we termed the latter as Ted, for Thermoproteales exchange of DNA system, with the CedA-like, CedA1-like, and VirB4-like components as TedA, TedC, and TedB, respectively. CedA2 is not conserved in the Ced systems from different species[26], and homologs or even counterparts of this protein are not identifiable in the Ted system.

Genomic loci containing the Ted system also commonly include genes encoding homologs of HerA helicase and MinD/ParA family ATPases, whereas *Pyrobaculum* and *Vulcanisaeta* species in addition carry the *rad50* recombinase genes. HerA helicase and Rad50 recombinase play an essential role during homologous recombination in hyperthermophilic archaea[43,44]. The co-localization of these genes with the Ted system suggests a coordination of the DNA import in Thermoproteales and DNA repair by homologous recombination. In the *A. tumefaciens* systems, MinD/ParA family ATPase, known as VirC1, spatially coordinates early conjugative DNA transfer reactions[45]. The finding that archaeal Ced and Ted systems form pili suggests that DNA transfer through these systems might be more similar to bacterial plasmid-mediated conjugation than previously recognized.

### Archaeal conjugation pili are stoichiometric complexes of pilins and lipids

With the atomic models for the pilin subunits docked within the respective cryo-EM maps, we observed unaccounted-for densities between each of the proteins in both maps. These densities were similar to the densities for lipid molecules found in each of the previously reported bacterial conjugation pili, where there is a stoichiometric 1:1 ratio of pilin:phospholipid[46,47]. However, the putative lipid densities in the archaeal conjugation pili were larger. It has been impossible to do the lipidomics analysis for the archaeal conjugation pili, due to our inability at this point to obtain a highly enriched preparation containing just the conjugation pili. However, given that *Aeropyrum* and *Pyrobaculum* contain only one membrane, the lipids in the pili must come from the archaeal cytoplasmic membrane.

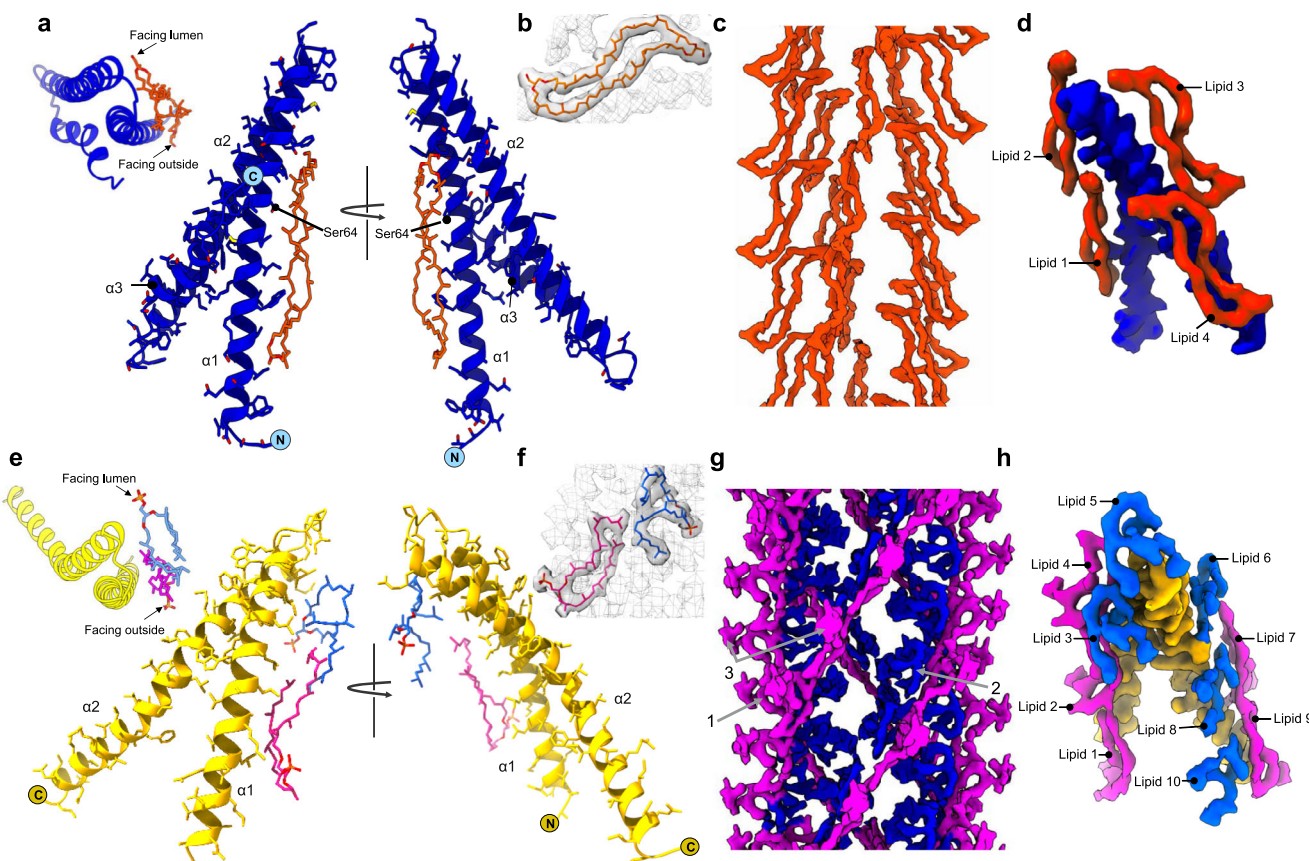

**Fig. 2 | Intricate pilin-lipid interaction networks within archaeal conjugative pili. a** Front and back views of a single asymmetric unit of the *P. calidifontis* pilus (blue) containing one pilin (TedC) and one phospholipid GDGT-0 (orange). One polar headgroup of the tetraether lipid faces the lumen and the other faces the outside of the pilus. A top view of a subunit, looking down the helical axis, is shown on the left for orientation. The isoprenoid chains of the GDGT-0 lipids are buried between hydrophobic helices and interact closely with helix α1 of the pilin shown but will also interact with α2 of neighboring pilins. **b** An atomic model of the cyclic GDGT-0 lipid docked within the lipid density. **c** Lipid density from *P. calidifontis* (blue). The density is very well resolved and shows that the GDGT-0 lipids have one head group facing the outside of the filament and the other head group facing the lumen. **d** A single pilin (blue) contacts four surrounding GDGT-0 lipids (orange). **e** The front and back view of the CedA1 pilin (yellow) of *A. pernix*. The asymmetric unit contains two lipids and one pilin, with the lipid in two different conformations: one having a partially folded shape (blue) and the other a crescent-like shape

(magenta). The crescent head group is facing the outside of the pilus while the isoprenoid chains are buried between the pilin subunits. The partially folded lipid's phosphate head group is facing the lumen of the pilus and the isoprenoid chains are bent and buried between the subunits. A top view of a subunit, looking down the helical axis, is shown on the left for orientation. For both lipids the contacts with the pilin are mediated by hydrophobic residues such as leucine, isoleucine and valine from helices α1 and α2. The crescent-like lipid has one hydrophobic interaction with the partially folded lipid. **f** Atomic models for the crescent-shaped and partially folded lipid docked into the lipid densities. **g** Lipid density from *A. pernix* (yellow). The lipid density is less resolved than in *P. calidifontis* but shows two C25-C25-diether lipids, one of which forms a crescent-like shape (arrow 1) and the other forms a partially folded shape (arrow 2). Both lipids are capped with extra density (arrow 3) which is likely a dihexose sugar attached to the phosphate head group. **h** A single protein subunit (yellow) makes contacts with ten lipids (five crescent-shaped (pink), and five partially folded (blue)).

In *P. calidifontis* the extended density could only be explained by a bipolar cyclic tetraether lipid (Fig. 2a–d). Bipolar archaeal lipids were proposed almost 40 years ago[48], but, to the best of our knowledge, they have never been directly visualized. The cryo-EM map was good enough to identify the cyclic lipid as a glycerol dialkyl glycerol tetraether species (GDGT), a dominant membrane lipid in many hyperthermophilic archaea, including *Pyrobaculum* species[49], and we have modeled the simplest form, GDGT-0, into the cryo-EM density (Fig. 2b). It is important to note the possibility that more complex forms of GDGT (GDGT-1 through 8) containing cyclopentane rings may fit into the density as well. Both head groups are found to be solvent-exposed, with one of the polar head groups facing the lumen and the other facing the outside of the pilus, while the acyl chains are buried between the hydrophobic helices of the pilin subunits (Fig. 2a). The GDGT-0 lipid is found positioned in the middle relative to helix α1 (Fig. 2a). Most of the pilin-lipid interaction network in the *P. calidifontis* pilus relies extensively on contacts with helix α1 in the asymmetric unit (ASU), but the lipid is sandwiched between α1 and α2 of a subunit in the

neighboring ASU. All contacts are primarily associated with hydrophobic residues apart from one polar electrically neutral serine residue, Ser64 (Fig. 2a). There appears to be minimal contact with the charged head groups. Thus, the protein-lipid interaction stabilization relies heavily on hydrophobic interactions for *P. calidifontis*. While the ASU consists of one protein to one lipid, there is a total of four lipids that make contact with the protein subunit TedC (Fig. 2d).

Interestingly, the lipid density was less resolved for *A. pernix* compared to the lipid density for *P. calidifontis*, even though the overall resolution for the *A. pernix* map was higher, 3.3 Å vs 4.0 Å, as determined by the map:map FSC (Table S1). In *A. pernix*, there are two lipids in every ASU, one of which adopts a partially folded conformation, with one of the isoprenoid chains folding back on itself, while the other has a crescent-like shape (Fig. 2e–g). Unlike most other members of the phylum Thermoproteota, the membranes of *Aeropyrum* species contain only a small amount of GDGT lipids. Instead, an unusual C25, C25-diether with a phospho-dihexose head group was the main lipid species observed by mass spectrometry of a

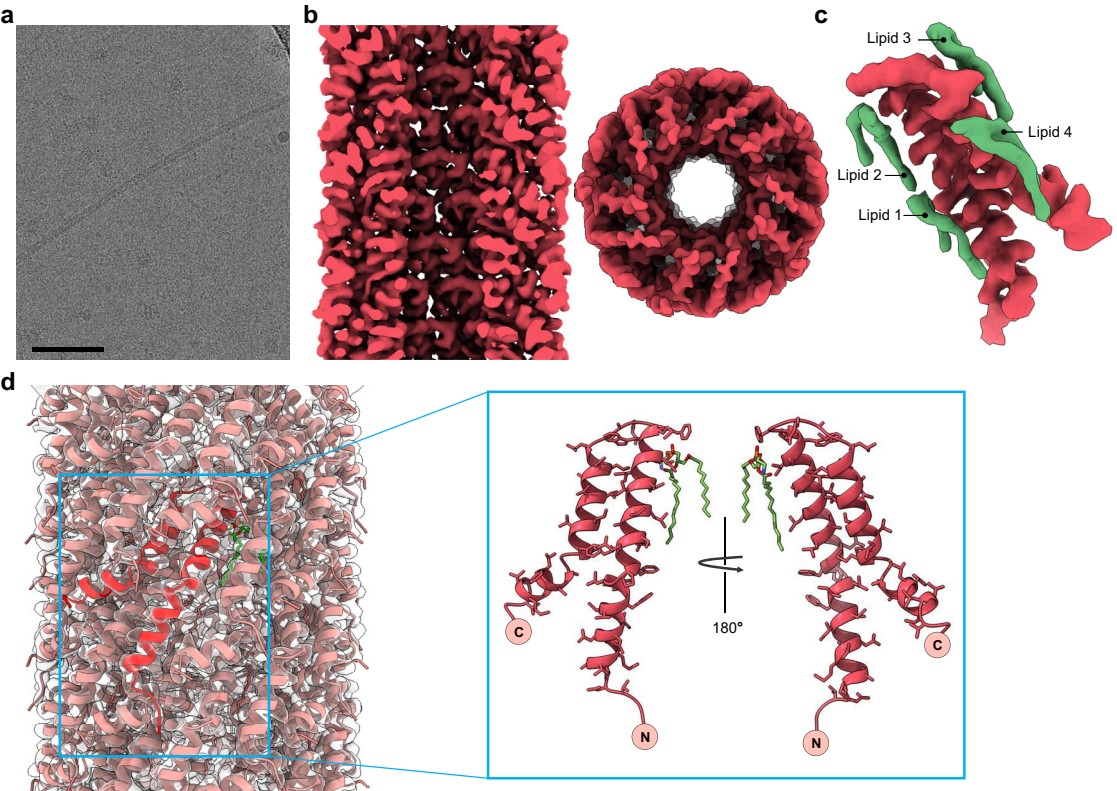

**Fig. 3 | Structure of the *A. tumefaciens* T-pilus. a** Representative cryo-electron micrograph of *A. tumefaciens* T-pilus, from 8,363 images collected. Scale bar, 50 nm. **b** Side and top view of the T-pilus cryo-EM density map at a resolution of 3.5 Å. The front half of the filament has been removed in the side view, and we are looking at the lumen. **c** While there is a 1:1 stoichiometry of lipids to pilins in the filament, a pilin (red) makes contact with four lipids (green). **d** Atomic model of the T-pilus in ribbon representation docked within the transparent cryo-EM density map. A single subunit model is shown in red. An inset on the right shows front and back views of the VirB2 subunit.

cellular membrane preparation of *A. pernix* (Supplementary Fig. 3a–b), consistent with the previous identification of this lipid in *A. pernix*[50]. Both non-protein densities in the *A. pernix* map could be fit with this lipid, suggesting that the same lipid is present in two different conformations (Fig. 2f). The crescent-shaped lipid head group is directed toward the extracellular space with the isoprenoid chains extended and buried between pilin subunits (Fig. 2e). The crescent-shaped lipid primarily contacts helix α1 of the pilin, with one potential hydrophobic interaction with the partially folded lipid (Fig. 2e). The contacts with helix α1 are mediated by leucine-rich hydrophobic interactions and one charged interaction between the phosphate head group and the positively charged Lys15. The second lipid density resulted in a model with bent isoprenoid chains and the phosphate head group directed toward the lumen (Fig. 2e). The isoprenoid chains are buried between the subunits. Contacts with the pilin are made exclusively by the hydrophobic residues leucine, iso-leucine, and valine from helices α1 and α2 (Fig. 2e). There are no observed contacts between the pilin and the lumen-facing phosphate head group of the second lipid. There are two lipids and one protein subunit in the ASU of the *A. pernix* CedA1 pilus, generating a complex network of lipid contacts for each protein subunit (Fig. 2g–h). In total, each CedA1 subunit will make contacts with 10 lipid molecules (Fig. 2h).

In addition to the lipids, weak peripheral density was observed for both of the archaeal conjugation pili (Supplementary Fig. 4) which would be consistent with glycosylation. However, this density was diffuse, and we could not see clear additional density on specific residues such as serine, threonine, or glutamine that might be targets of such glycosylation[51].

## Conjugation pili of *Agrobacterium tumefaciens*

To extend the comparison between the archaeal mating pili and the existing structures of bacterial ones, we used cryo-EM to solve the structure of the *A. tumefaciens* T-pilus encoded by the pTiC58 plasmid to 3.5 Å resolution (Fig. 3a, b; Table S1). The T-pilus has a fivefold rotational symmetry with a rise of 13.7 Å and a twist of 32.5° per sub-unit. Similar to other bacterial conjugation pili[46,47], there is a stoichio-metry of one lipid molecule to each protein subunit. However, unlike in pED208[46] and pKpQIL[47], where each protein subunit contacts five lipid molecules, in T-pilus, each pilin subunit contacts four lipid molecules (Fig. 3c), in a fashion similar to the TedC pilins (Fig. 2d).

We conducted comprehensive shotgun lipidomics by electro-spray ionization mass spectrometry (ESI-MS) on isolated pili to identify and detail the lipid species tightly associated with fibrillar pili-forming proteins. The isolated pili were pre-treated with PLA2 to hydrolyze all (contaminating) phospholipids that were not stably associated with pili proteins. As a control, we analyzed isolated pili from pED208, whose associated lipids were previously reported as being two sub-species of the anionic lipid phosphatidyl-glycerol (PG) (Costa et al., 2016). Our analysis (Supplementary Fig. 5) confirmed and extended these results, showing that these two species (PG 16:0/16:1 and PG 16:0/18:1) are indeed the most abundant lipids associated with pED208, and that three other similar PG species are also present. Thus, ESI-MS allows quantitative analysis of lipid species associated with bacterial pili. The same analysis applied to VirB2 T-pili (Supplementary Fig. 5) revealed a strikingly different set of PLA2-resistant lipids, with >65% of the phospholipids being phosphatidyl-ethanolamine (PE). Interest-ingly, the acyl chains of T-pilus lipids (PE 16:0/16:1 and PE 16:0/18:1) were similar to those observed in pED208, despite having different

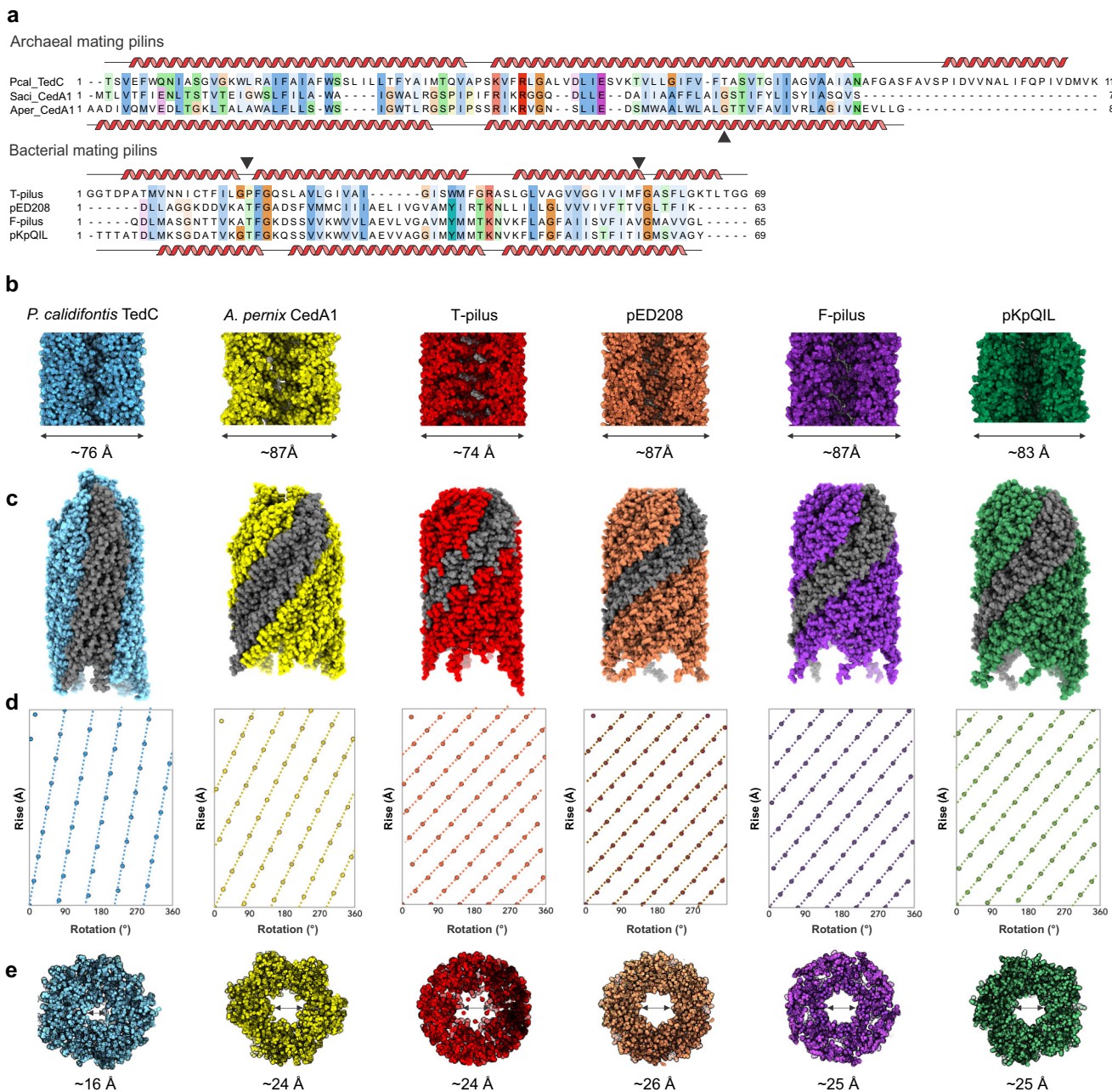

**Fig. 4 | Comparison of archaeal and bacterial conjugative pili. a** Sequence alignments of archaeal (top) and bacterial (bottom) mating pili. Sequences of mature pilins are shown with the secondary structure elements determined from the structure for the pilins shown above and below the corresponding sequences. Kinks in the α-helices are indicated with triangles. **b, e** Comparison of the lumen and outer diameter of archaeal *P. calidifontis* (blue) and *A. pernix* (yellow) with bacterial pili: *A. tumefaciens* T-pilus (red), pED208 (mauve), F-pilus (purple) and pKpQIL (green). The outer diameters range between 74 Å to 87 Å. The lumen diameters range between 16 Å to 26 Å, but in some cases cannot be easily reduced to a single number. **c, d** Atomic models with a single strand, shown in gray, show connectivity between the subunits. This connectivity is also represented with helical nets for the archaeal and bacterial pili. The helical nets show the unrolled surface lattice viewed from the outside of the filament. Each point represents a subunit, and the dotted lines are drawn to highlight the fact that all have right-handed 5-start helices. All of the pili have substantial connectivity between subunits along these 5-start helices.

chemistry of the headgroup. A minor fraction (<20%) of PLA2-resistant lipids were phosphatidyl-cholines (PC).

It is important to emphasize the essential need for PLA2-catalyzed hydrolysis of contaminating phospholipids in these assays, as the pili-bound lipids are relatively unabundant, and even minor membrane contamination will overwhelm the signal. We, therefore, expect that in the absence of phospholipase treatment, any lipidomics analysis of the T-pilus will mainly find the lipids present in the contaminating membrane blebs and vesicles. This might explain the large difference between our results, where PE is the dominant lipid in the pili, and that

found in two other very recent reports, where either PC[52] or PG[53] were found as the dominant lipid in the T-pilus. In neither of these two studies was phospholipase treatment used to minimize the contribution of contaminant lipids.

The 3.5 Å resolution of the T-pilus map allowed for unambiguous model building and interpretation of the protein subunit, VirB2 (Fig. 3d). The atomic model of VirB2 shows clear structural homology to the TraA subunit of the F-pilus (Supplementary Fig. 1c). One clear difference between VirB2 on the one hand, and TraA of pED208 on the other, is that there are kinks in helix 1 and 3 of the VirB2 subunit,

produced by Pro23 and Phe61, respectively (Supplementary Fig. 1c). The electrostatic surface for the lumen of the T-pilus generated by the atomic model is overall quite positive. The headgroup of PE is zwitterionic, with a net neutral charge, and the addition of the PE lipid into the T-pilus model does not significantly change this electrostatic surface, which results in a more positively charged lumen compared to the lumen of pED208[46] (Supplementary Fig. 6). Interestingly, when viewed from the top there is an alternating positive to a negative charge arising from the arginine and the lipid, respectively (Supplementary Fig. 6). The T-pilus is known for its diverse substrates, enabling transport of both ssDNA-VirE2[54] as well as effector proteins[55] whose transport is independent of the DNA[30]. The inclusion of VirE2 is thought to protect the T-strand from nuclease degradation in *A. tumefaciens* and facilitate its transfer through the T4SS and T-pilus[54]. An AlphaFold prediction for the structure of VirE2 suggests that it would be too large to pass through the lumen of the T-pilus in a natively-folded state. We, therefore, suggest that it must be partially unfolded to allow for such transport. While the lumen of other bacterial mating pili may have evolved to be optimal for DNA transport, the lumen of the T-pilus, also used for the transport of a diverse range of other substrates, has a positive electrostatic potential that would still allow DNA transfer, but not be optimal due to the greater friction resulting from DNA sticking to the walls. Obviously, this would suggest that the other substrates are likely to have overall positive electrostatic surfaces, and it has previously been noted that the effector proteins, where characterized, carry C-terminal domains that are positively charged[30].

The high resilience of the T-pilus to extreme chemical or physical conditions has been reported in a previous study[37], and its stability was attributed to the putative cyclic nature of the T-pilin[35,36]. Surprisingly, the cryo-EM structure of VirB2 reveals no cyclization of the pilin. Similar to other bacterial conjugative pili and TedC of *P. calidifontis*, VirB2 is proteolytically processed by a signal peptidase[56]. In the mature pilin, residues QSAG from the N-terminus and G from the C-terminus are not seen in the density map, most likely due to disorder, and have not been built into the atomic model. The two residues suggested to be covalently linked in the T-pilus[35] were Gln1 and Gln74 (using our numbering for the mature pilin), neither of which are in our atomic model. But the five missing residues would be unable to span the distance of ~34 Å between Gly5 and Gly73 (the first N-terminal residue and last C-terminal residue in the model), so it is not possible that the subunit really is cyclic. This leads us to the question of what is the basis for the resiliency of the T-pilus? To answer this question, we subjected the T-pilus to many of the same extreme chemical or physical conditions reported by Lai and Kado[37] (Supplementary Fig. 7, Table S2) and compared the resiliency to the well-studied F-pilus which is known to be non-cyclic. Interestingly, the T-pili look intact under 50% glycerol and 4 M urea, flexuous and partially degraded under high temperatures (70 °C) and 0.1% SDS, and depolymerized under 1% Triton-X-100. In contrast, the F-pili is much more stable and remain intact under all conditions, including 1% Triton-X-100 (Supplementary Fig. 7 and Table S2). These results suggest that the architecture of bacterial conjugation pili is generally very stable and not related to cyclization. Further, we see no evidence of extensive glycosylation for the T-pilus (Supplementary Fig. 4), nor was any potential glycosylation described for previous conjugation pili structures[46,47], so we can exclude extensive glycosylation which has been suggested as a mechanism for stabilizing extracellular archaeal filaments in the most extreme environments[51].

**The lumen of prokaryotic pili is too narrow to transfer dsDNA**

With structures for four bacterial conjugation pili (pED208, F, pKpQIL, and pTiC58 from *A. tumefaciens*) and two archaeal ones (*A. pernix* and *P. calidifontis*), it is clear that there is a common architecture for all and obvious homology, despite negligible sequence similarity (Fig. 4a). Each prokaryotic pilin subunit consists of two or three hydrophobic

α-helices, with kinks appearing in some of the helices, such as in *A. pernix* and *A. tumefaciens* (Fig. 2a, d, 3d and 4a). When a heat-map of global structural similarity (based upon the Dali server[57] Z-scores) between a single pilin from all known structures of prokaryotic conjugation pili was generated, unsurprisingly a clustering of the archaeal pilins was observed (Supplementary Fig. 8). Notably, however, the mature *P. calidifontis* pilin contains an additional hydrophobic α-helix (α3) compared to pilins encoded by *A. pernix* and bacterial plasmids. Despite the *A. tumefaciens* pilin having high structural similarity with pED208 and pKpQIL, it was clustered with the archaeal pilins rather than bacterial pilins from the F-pilus, pED208, and pKpQIL, which formed a separate cluster. The clustering of *P. calidifontis*, *A. pernix*, and *A. tumefaciens* pilins is likely due to the helix kinks which are absent in the other bacterial pilins.

We compared the external diameters of the archaeal and bacterial (T-pilus, pED208, F-pilus, and pKpQIL) conjugation pili as well as the diameters of their central channels, which allow the transfer of DNA (Fig. 4b, e). It must be noted that all such measurements are quite approximate, as discussed recently[58], and usually ignore the contribution of hydrogens and tightly bound water molecules. Further, structural varicosity in both archaeal pili complicates reducing the diameter to a single number. Nonetheless, with unavoidable approximations, lumen diameters (Fig. 4b, e), range from 16 Å to 26 Å. The outer diameter of the *A. pernix* pilus more closely resembles the outer diameters of the bacterial F and F-like pili, while *P. calidifontis* has an outer diameter that is approximately the same as the T-pilus. The lumen diameter of *A. pernix* pilus is very similar to that of the T-pilus, whereas the lumen diameter of *P. calidifontis* is considerably narrower (Fig. 4e).

To determine whether dsDNA could pass through the lumen of the prokaryotic conjugative pili, we placed a model for B-form dsDNA within the lumen of the conjugation pili models from *P. calidifontis*, *A. pernix*, and *A. tumefaciens*. We observed extensive clashes between dsDNA and the atomic surface of all models (Supplementary Fig. 9). These observations suggest that ssDNA, not dsDNA, passes through conjugative pili, although no experimental evidence exists to support this notion. Indeed, using synthetic nanopores in ultrathin silicon nitride membranes, it has been shown that ssDNA permeates pores with diameters as small as 10 Å[59]. Hence, pore diameters of all prokaryotic conjugative pili are sufficiently wide for the passage of ssDNA, but not dsDNA. Previous structural studies have suggested that the lumen of the F-pilus has an overall negative charge which contributes to a repulsive force that will keep negatively charged ssDNA away from the wall of the lumen, effectively lowering friction[46,47]. In *A. pernix* each pilin is bound to two diether lipids with phosphorylated dihexose (i.e., glucose-inositol) head groups. One of these lipids faces the lumen of the central channel where it might also provide a similar negative charge to facilitate DNA transfer (Fig. 2e).

**Different helical symmetries still allow quasi-equivalent interactions**

The twist of the archaeal pili from both *P. calidifontis* (74.2°) and *A. pernix* (76.5°) is similar to that of the pKpQIL pilus from *K. pneumoniae* (77.6°), generating strong connectivity along 5-start protofilaments (Fig. 4c–d). Similar 5-start protofilaments are observed in other bacterial conjugation pili as well, including the *A. tumefaciens* T-pilus reported here, the pED208 F-like pilus[46], and the F-pilus[46]. However, those pili have a C5 rotational symmetry rather than the 1-start helical symmetry observed in pKpQIL[47], *P. calidifontis*, and *A. pernix* (Fig. 4c, d). The difference in helical symmetries results in very small differences in intermolecular interfaces, much like what is observed for pED208 and pKpQIL, which have C5 and C1 symmetries, respectively[47]. This is similar to the quasi-equivalence phenomenon reported in other helical tubes made of helix-turn-helix subunits, such as in archaeal virus SMV1[60], where very similar interfaces can be preserved even

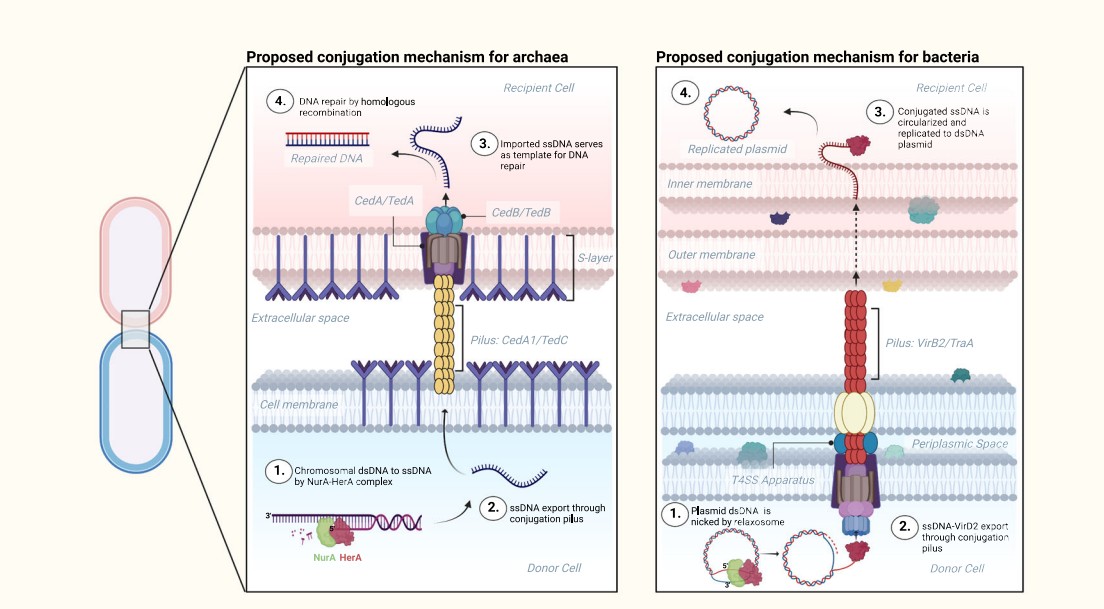

**Fig. 5 | Proposed conjugation mechanisms between donor and recipient cells in archaea (left) and bacteria (right).** The schematic shows how ssDNA substrates are generated by the HerA-NurA machinery in the donor archaeal cells and by the plasmid-encoded relaxosome in bacteria. Note that CedA and, potentially, Ted systems function as DNA importers rather than DNA exporters, contrary to the bacterial plasmid conjugative machinery.

though there are large apparent changes in symmetry (between C7 and C1 in SMV1).

## Discussion

Bacterial conjugation machinery is nearly exclusively encoded by plasmids or integrating and conjugating elements and hence typically promote the transfer of the cognate mobile genetic elements, with the occasional transfer of the host DNA, e.g., when the plasmid origin of transfer (*oriT*) is recombined into the cellular chromosome[61]. An exception to this paradigm is the conjugation-like process, dubbed distributive conjugal transfer, described in *Mycobacterium* species, whereby large fragments of the chromosomal DNA are exchanged between the donor and recipient species[62]. However, although superficially this form of DNA transfer is reminiscent of the classical plasmid conjugation, it is mediated by a poorly understood mechanism involving the Type VII secretion system, rather than the T4SS[63]. Here we show that the Ced system which imports DNA in hyperthermophilic archaea of the order Sulfolobales[26], and the related Ted system of Thermoproteales described herein, are domesticated derivatives of the T4SS. Both Ced and Ted systems encode homologs of the VirB4 ATPase (CedB/TedB), VirB6 membrane pore (CedA/TedA), and VirB2 conjugative pilin (CedA1/TedC). Nevertheless, the individual components display no recognizable sequence similarity between the Ced and Ted systems, and the assembly pathways might also differ. For instance, whereas the pilins in the Ced system are apparently secreted without processing, the N-terminal signal sequence of TedC appears to be cleaved by a signal peptidase I. In this respect, TedC is more similar to the bacterial plasmid VirB2-like pilins. It is interesting to note that the N-terminal region of CedA proteins is homologous to nearly entirety of the CedA1 pilin (Supplementary Fig. 2c), suggesting a common export pathway for CedA and CedA1 and that one has evolved from the other. Notably, the same result was obtained when VirB2 and VirB6 of *A. tumefaciens* were compared, albeit with a lower significance score (Supplementary Fig. 2d). It is not possible to claim with confidence which of the two proteins is ancestral. However, given the central role of the VirB6-like pore for conjugation and the absence of conjugative pili in monoderm bacteria, it is tempting to suggest that the pilin is a more recent addition to the T4SS apparatus to facilitate the DNA transfer between spatially separated cells. According to this scenario, the stoichiometric incorporation of lipids, a unique feature of bacterial conjugation pili and their archaeal homologs, might be a vestige of the ancestral function of these pilins as bona fide membrane proteins. Notably, our results provide the first direct visualization of archaeal GDGT lipids in the *P. calidifontis* pilus and highlight the flexibility of diether lipids in the pilus of *A. pernix*. The pilins are surrounded by lipid molecules, with most interactions holding the pili together being between pilin subunits and lipids. Thus, in a way, the conjugative pili can be regarded as highly ordered extensions of the cytoplasmic membrane.

Given the direct visualization of horizontal gene transfer between spatially separated bacterial cells using fluorescence microscopy[64], it is clear that bacterial conjugation pili can act as conduits for DNA transfer. However, the possibility still exists that the main role of such pili is to depolymerize and bring two mating cells into physical juxtaposition, and that the bulk of DNA transfer only takes place when this conjugation junction is established. Our structural results cannot address this possibility. Many of the proteins involved in conjugation have been identified in bacteria, such as in *E. coli*[65,66] and *A. tumefaciens*[67], but these proteins have remained elusive in archaea. Mutagenesis studies in bacteria have shown ssDNA to be the genetic material exported from the donor to the recipient cell via the conjugation pilus[64,68], where dsDNA is nicked by the enzyme relaxase. Relaxase in complex with several other proteins, known collectively as the relaxosome, is responsible for mediating the unwinding of ssDNA. To our knowledge, there are no apparent homologs of bacterial relaxases encoded in the archaeal genomes or conjugative plasmids[23]. Nevertheless, our data indicate that archaeal pili, similar to their bacterial counterparts, most likely transfer ssDNA. The observation that Ted genes in some genera of Thermoproteales co-occur with genes encoding Rad50 and HerA-like helicase, might hold a clue to this conundrum. In hyperthermophilic archaea, *herA* and *rad50* usually form an operon with the genes encoding nuclease NurA and Mre11, and the four proteins function during DNA damage repair through homologous recombination[69]. NurA, an RNase H-fold nuclease, is endowed with the endonuclease and exonuclease activities that are modulated by the HerA helicase[70,71]. The integrated activity of NurA-

HerA is responsible for DNA end-resection, a process that generates the 3′ single-stranded tails that are subsequently coated by the Rad50 recombinase to initiate strand invasion and DNA repair. In Sulfolobales, expression of the HerA operon, Ced, and Ups systems is activated by DNA damage and all three operons are coregulated by the transcription factor B3[72]. We hypothesize that the ssDNA substrate for the transfer through Ced and Ted systems is generated by the activity of NurA-HerA system, rather than by a dedicated relaxosome as in the case of bacterial plasmid conjugation systems (Fig. 5). Notably, Ted system, but not Ced, apparently includes a ParA/MinD-family ATPase related to the VirC1 protein of *A. tumefaciens* plasmid Ti, which functions during the delivery of the relaxosome-bound ssDNA to the T4SS complex[45], and a similar role can be postulated for the homologous protein of the Ted system. The narrower pore in the *P. calidifontis* mating pilus, compared to *A. pernix* and the bacterial mating pili (Fig. 4, Supplementary Fig. 9), may reflect the fact that only ssDNA is transferred, rather than a relaxase-ssDNA complex.

Our current study provides new insights into the mechanism of horizontal DNA transfer in hyperthermophilic archaea through the domesticated conjugative T4SS apparatus. Such domestication is a remarkable example of the 'guns-for-hire' paradigm[73], whereby molecular machines evolving at the interface of the interaction between mobile genetic elements and their hosts are captured and repurposed by the competing parties. To our knowledge, the domestication of the conjugative apparatus for DNA transfer has not been reported in other organisms. Many questions remain unanswered, including the generation of ssDNA substrates for intercellular transfer as well as the mechanistic details of the biogenesis and full molecular complexity of the Ced and Ted systems. Notably, none of the species in the orders Thermoproteales and Desulfurococcales, including members of the genera *Pyrobaculum* and *Aeropyrum*, are genetically tractable. Thus, our present study further highlights the utility of cryo-EM in gaining important insights into the biology of non-model organisms.

## Methods

### Cultivation of archaeal cells and preparation of pili samples

*Pyrobaculum calidifontis* DSM 21063[40] and *Aeropyrum pernix* K1 DSM 11879[38] cells were purchased from the DSMZ culture collection. *P. calidifontis* was grown in 1090 medium (1.0% tryptone, 0.1% yeast extract, 0.3% sodium thiosulfate, pH 7) at 90 °C without agitation. Pre-culture (30 mL) was started from a 200 μL cryo-stock, grown for 2 days and then diluted into 200 mL of fresh medium. When $OD_{600}$ reached ~0.2, the cells were collected by centrifugation (Sorval SLA1500 rotor, $7438 \times g$, 10 min, 20 °C). The resultant pellet was re-suspended in 10 mL of phosphate-buffered saline (PBS) buffer, and the cell suspension was vortexed for 15 min to shear off the extracellular filaments. The cells were removed by centrifugation (Eppendorf F-35-6-30 rotor, $7197 \times g$, 20 min, 20 °C). The supernatant was collected and the filaments were pelleted by ultracentrifugation (Beckman SW60Ti rotor, $194,038 \times g$, 2 h, 15 °C). After the run, the supernatant was removed and the pellet was re-suspended in 200 μL of PBS buffer.

*A. pernix* K1 cells were grown in 3ST medium (35 g/L Sea salts [Sigma], 0.1% tryptone, 0.1% yeast extract, 0.1% thiosulfate, pH 7) at 90 °C without agitation. Pre-culture (10 mL) was started from a 1 mL cryo-stock, grown for 3 days, and then diluted into 60 mL of fresh 3ST medium. After 3 days of growth 100 mL of fresh media was added to the culture and the growth was continued for another 3 days. Then the cells were removed by centrifugation (Sorval SLA1500 rotor, $7438 \times g$, 10 min, 20 °C), and the filaments were pelleted from the supernatant by ultracentrifugation (Beckman SW60Ti rotor, $194,038 \times g$, 2 h, 15 °C) and re-suspended in 200 μL of PBS.

### pED208 pili purification

The F-pili were purified as described in Costa et al., 2016 with some minor modifications. In short, *E. coli* JE2571 harboring the pED208-

plasmid were grown on large (24 × 24 cm) LB agar plates overnight. Bacteria were gently collected from the plates with SSC buffer (15 mM sodium citrate pH 7.2 150 mM NaCl) and left to resuspend for 2 h at 4 °C with mixing, followed by two rounds of centrifugations at $10,800 \times g$ for 20 min. The pili were precipitated from the supernatant by addition of 500 mM NaCl and 5% PEG 6000, followed by incubation for 2 h at 4 °C. Precipitate was rescued by centrifugation at $15,000 \times g$ for 20 min and re-suspended in 120 ml of water, followed by another round of precipitation as described above but this time the precipitate was re-suspended in 1 ml of PBS (pH 7.4) buffer. The suspension was layered on pre-formed CsCl step gradients (1.0–1.3 g/cm³) and separated by 17 h centrifugation at $192,000 \times g$ in 4 °C. The gradient was fractionated and the fraction containing the F-pili was dialyzed against PBS (pH 7.4). Purity of the F-pilus was assessed by SDS-PAGE and the presence of the F-pilin (TraA) was further confirmed by mass spectrometry.

### *A. tumefaciens* pili purification

Methods for *Agrobacterium tumefaciens* T-pilus isolation and concentration are adapted from Lai and Kado[74]. Briefly, *A. tumefaciens* C58ΔvisR (flagella-knockout) was streaked from a stab or frozen culture on LB agar with no antibiotics at 28 °C. After colonies appeared, about 2 days, a single colony was cultured overnight in 5 mL 523 media in a covered culture tube with shaking at 19 °C. The next day the turbid culture was transferred to a centrifuge tube and pelleted by centrifuging 15 mins at $119.5 \times g$. Media was aspirated from the culture tube and the pellet was gently re-suspended in 25 mL AB/MES with 5% glucose, then incubated at 19 °C for 4 hours. The culture was then spread on 20 petri dishes or six screening trays containing AB/MES with 5% glucose, 200 μM acetosyringone to induce pili growth, 1.2% agar. Agar plates were incubated at 19 °C for 6 days. After a lawn developed on each plate, 1 mL of cold 10 mM sodium phosphate buffer, pH 5.3 buffer was added to each petri dish (or 5 mL for screening trays) and layers of bacteria were scraped off with a cell spreader and transferred to a 50 ml Falcon tube on ice. This step was repeated with a second aliquot of buffer added to remove the remaining bacteria. The total volume of bacteria and buffer was about 50 mL. The bacterial suspension was pipetted up and down to break up the biofilm then gently pushed through 1 mL of glass wool in a 30 mL syringe to strain out agar gel. The strained bacterial suspension was then forced through a 26-gauge needle eight times total to shear the T-pili off of the bacteria. After shearing, the suspension was transferred to a 0.2 μm disposable filter vacuum flask and filtered on ice to separate sheared pili from whole cells, periodically removing build-up from the filter. The isolated pili were concentrated by a factor of 10 using a 100 MWCO centrifugal filter. The pili were rinsed twice with 10 mM Tris-HCl, 100 mM NaCl, pH 7.5 buffer, re-suspended to 10% of the starting volume, and frozen at −80 °C for storage.

### Cryo-EM sample preparation and data collection

A 3 μL aliquot of sample containing either *Pyrobaculum calidifontis*, *Aeropyrum pernix*, or *Agrobacterium tumefaciens* pili was applied to a plasma cleaned (Gatan Solarus) lacey carbon grid (Ted Pella, Inc.), blotted with automated blotting for 3 s at 90% humidity and flash frozen in liquid ethane using an EM GP Plunge Freezer (Leica). The dataset used for structure determination was collected at the Molecular Electron Microscopy Core at the University of Virginia on a Titan Krios EM operated at 300 keV, equipped with an energy filter and K3 direct electron detector (Gatan). An energy filter slit width of 10 eV was used during data collection and was aligned automatically every hour. All 8127 *P. calidifontis*, 598 *A. pernix*, and 8363 *A. tumefaciens* movies were collected in counting mode using EPU v2.4 (Thermo Fisher) at a magnification of 81 K, pixel size of 1.08 Å, and a defocus range from −2.2 to −1.2 μm. Data were collected using a total dose of 50 e⁻ Å⁻² across 40 frames with an exposure time of 2.98 s.

## Data processing and helical reconstruction

Unless otherwise stated, all data processing was done using cryoS-PARC v3.2.0[75]. Movies were corrected for full-frame motion using patch motion correction followed by patch CTF Estimation[76]. After CTF estimation, micrographs were sorted and selected based on estimated resolution (0 to 4 Å), defocus (0.6 to −2.6 μm), ice thickness, and total full-frame motion. Initial particles were automatically picked using 'Filament Tracer' with a filament diameter of 100–160 Å and an overlap fraction of 0.05–0.07. Particles were extracted at a box size of 300 or 320 pixels, followed by 2D classification. Class averages containing filaments distinguishable from noise were selected for template-based particle picking. A total of 549,015 and 427,344 filament segments were extracted using a box size of 320 Å, for *P. calidifontis* and *A. pernix*, respectively. A total of 197,531 T-pilus filament segments were extracted using a box size of 300 Å for *A. tumefaciens*. These particles were sorted using two iterative rounds of 2D classification with 50 classes each, number of online-EM iterations set to 20, and a batch size of 100 per class. The final iteration of 2D classification yielded a subset of 71,981, 44,262, and 49,308 filament segments for *P. calidifontis*, *A. pernix*, and *A. tumefaciens*, respectively. Reconstructions of archaeal and prokaryotic conjugation pili were generated using the following method: (1) an averaged power spectrum was generated using the raw images of aligned filament segments selected from 2D classification, (2) layer lines were indexed to produce a list of possible helical symmetries, and (3) the correct helical symmetry was determined by trial and error by inspection of an output 3D map looking for obvious structural motifs (i.e., recognizable secondary structural and amino acid side-chain densities). Particles were further refined using local CTF refinement, and another round of helical refinement was performed to generate the final reconstruction. The final resolution achieved for *P. calidifontis*, *A. pernix*, and *A. tumefaciens*, were 4.0 Å, 3.3 Å, and 3.5 Å, respectively. The cryo-EM and refinement statistics for each conjugation pilus are listed in Table S1.

## Model building and refinement

The sequence identity of the subunit for *P. calidifontis* and *A. pernix* was unknown. Using AlphaFold[77] the sequence identity for *A. pernix* pilin was narrowed to the one that best fit the cryo-EM density map. Using DeepTracer-ID[39] we were able to determine the pilin identity of *P. calidifontis* directly from the cryo-EM map. With the sequences identified for their respective map the side chains of each ASU model were adjusted manually in COOT[78] and inspected using UCSF Chimera[79]. For the *A. tumefaciens* structure the density was good enough to trace the entire backbone and localize most side chains. The cryo-EM structure of pKpQIL (PDB ID: 7JSV) was used as a starting point for building the *A. tumefaciens* cryo-EM model. Following model completion, side chains of the model were manually adjusted in COOT[78] and inspected in UCSF Chimera (Pettersen et al., 2004). All models were refined using PHENIX real-space refinement[80]. Refinement included global minimization, B-factor optimization, and applied secondary structure and Ramachandran restraints. The final models were validated with the MolProbity[81] implementation in PHENIX. Refinement statistics for each filament are listed in Table S1. Both cryo-EM maps and atomic coordinates have been deposited with the Electron Microscopy Data Bank and Protein Data Bank with the accession codes given in Table S1. Model-map correlation coefficients were also used to estimate the resolution of the reconstructions and are listed in Table S1.

## Sequence analyses

Multiple sequence alignment of bacterial and archaeal pilins was made using PROMALS3D and manually adjusted[82]. Genomic neighborhoods were analyzed using the enzyme function initiative-genome neighborhood tool (EFI-GNT)[83]. Profile-profile comparisons and annotation of proteins encoded in the vicinity of TedC pilin in Thermoproteales

were performed using HH-suite package v3[84]. Profiles of the query sequences were constructed by running three iterations of HHblits against the UniRef90 database and the resultant profiles were compared against various profile databases, including the protein family (Pfam) database, the Protein Data Bank (PDB), and the NCBI Conserved Domain database, all available from http://ftp.tuebingen.mpg.de/pub/protevo/toolkit/databases/hhsuite_dbs/[85]. Signal peptides were predicted using SignalP v6[42].

## Mass spectrometry and lipidomics of *A. pernix*

The freeze-dried cell preparation was extracted using a modified Bligh-Dyer protocol[86]. Part of the extract was acid hydrolyzed by refluxing with 5% HCl in methanol for 3 hours to release core ether lipids. Both the Bligh-Dyer extract (containing the intact polar lipids with head groups) as well the hydrolyzed extract was analyzed by ultra-high performance liquid chromatography-high-resolution mass spectrometry using a Q Exactive Orbitrap MS following Bale et al.[86].

## Electron spray ionization and tandem MS-MS of pED208 F-pilus and pC58 T-pilus lipidomics

Lipidomics was performed at Lipotype, GmbH (Dresden, Germany) as described previously[87–90].

**Nomenclature.** The following lipid names and abbreviations are used. Cer – Ceramide, Chol – Cholesterol, CL – cardiolipin, DAG – Diacylglycerol, HexCer – Glucosyl/Galactosyl Ceramide, PA – Phosphatidic Acid, PC – Phosphatidylcholine, PE – Phosphatidyl-ethanolamine, PG – Phosphatidyl-glycerol, PI – Phosphatidylinositol, PS – Phosphatidylserine, and their respective lysospecies: lysoPA, lysoPC, lysoPE, lysoPI and lysoPS; and their ether derivatives: PC O-, PE O-, LPC O-, LPE O-; SE – Sterol Ester, SM – Sphingomyelin, TAG – Triacylglycerol.

Lipid species were annotated according to their molecular composition as follows: [lipid class]-[sum of carbon atoms in the fatty acids]:[sum of double bonds in the fatty acids];[sum of hydroxyl groups in the long chain base and the fatty acid moiety] (e.g., SM-32:2;1). Where available, individual fatty acid composition following the same rules is given in brackets (e.g., 18:1;0-24:2;0).

**Lipid extraction.** Samples were extracted and analyzed as described[87–91], which is a modification of a previously published method for shotgun lipidomics[88]. Briefly, samples were suspended in 150 μL of 150 mM ammonium bicarbonate in water and spiked with 20 μL of internal standard lipid mixture, then extracted with 750 μL chloroform/methanol 10:1 (v:v) mixture for 2 hours at 4 °C with 1400 rpm shaking. After centrifugation (3 min, 3000 × g) to facilitate phase partitioning, the lower, lipid-containing, organic phase was collected (1st step extract), and the remaining water phase was extracted further with 750 μL chloroform/methanol 2:1 (v:v) mixture under the same conditions. Again, the lower, organic phase was collected (2nd step extract). Extracts were dried in a speed vacuum concentrator. 120 μL of a dried 1st step extract underwent acetylation with 75 μL acetyl chloride/chloroform 1:2 (v:v) mixture for 1 h to derivatize cholesterol. After completing the reaction, the mixture was dried. 120 μL of a dried 1st step extract and a derivatized extract were re-suspended in an acquisition mixture with 8 mM ammonium acetate (400 mM ammonium acetate in methanol:chloroform:methanol:propan-2-ol, 1:7:14:28, v-v:v:v). 120 μL of the 2nd step extract was re-suspended in an acquisition mixture with 30 μL 33% methylamine in methanol, in 60 mL methanol:chloroform 1:5 (v:v). All liquid handling steps were performed using a Hamilton STARlet robotic platform.

**PLA2 treatment.** 0.2 U PLA2 was added to Eppendorf tubes containing samples with a concentration of ~18 μM of pili from either pED208 F-pilus or pC58 T-pilus. These samples were then incubated for 60 minutes at 37 °C and flash-frozen in liquid ethane.

**Lipid standards.** Synthetic lipid standards were purchased from Sigma-Aldrich (cholesterol D6), Larodan Fine Chemicals (DAG, TAG), and Avanti Polar Lipids (all remaining lipids). Standard lipid mixtures were chloroform/methanol 1:1 (v:v) solutions containing the lipids listed in Table S3.

## Lipid spectrum acquisition
Extracts in acquisition mixtures were infused with a robotic nanoflow ion source (TriVersa NanoMate; Advion Biosciences) into a mass spectrometer instrument (Q Exactive, Thermo Scientific). Cer, DiHex-Cer, HexCer, lysolipids, and SM were monitored by negative ion mode FT MS. PA, PC, PE, PI, CL, PS, and ether species were monitored by negative ion mode FT MS-MS. Acetylated cholesterol was monitored by positive ion mode FT MS. SE, DAG, TAG, and species were monitored by positive ion mode FT MS-MS.

## Lipid identification and quantification
Automated processing of acquired mass spectra, identification, and quantification of detected molecular lipid species was performed by LipidXplorer software[92]. Data post-processing and normalization were performed using an in-house developed data management system. Only lipid identifications with a signal-to-noise ratio >5, an absolute abundance of at least 1 pmol, and a signal intensity 5-fold higher than in corresponding blank samples were considered for further data analysis.

## Chemical and physical treatments of pED208 F-pilus and pC58 T-pilus
The following chemical or physical treatments were introduced and incubated at room temperature (23 °C) unless otherwise stated for ~10 minutes to samples of either pED208 F-pilus or pC58 T-pilus: 50% glycerol, high temperature (70 °C), 0.1% SDS, 1% Triton-X-100, and 4 M urea.

## Negative stain transmission electron microscopy of treated pED208 F-pilus and pTiC58 T-pilus
2 µL of the sample either of pED208 F-pilus or pTiC58 T-pilus subjected to their respective chemical or physical treatment were applied to a carbon grid and stained with 2% uranyl acetate and examined by transmission electron microscopy using a Tecnai T12 at 80 kV.

## Reporting summary
Further information on research design is available in the Nature Portfolio Reporting Summary linked to this article.

## Data availability
The atomic model for the *P. calidifontis* pilus was deposited in the Protein Data Bank with accession code 8DFT, and the corresponding map was deposited in the Electron Microscopy Data Bank with accession code EMD-27413. The atomic model for the *A. pernix* pilus was deposited in the Protein Data Bank with accession code 8DFU, and the corresponding map was deposited in the Electron Microscopy Data Bank with accession code EMD-27414. The atomic model for the *A. tumefaciens* pilus was deposited in the Protein Data Bank with accession code 8EXH, and the corresponding map was deposited in the Electron Microscopy Data Bank with accession code EMD-28657.

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

## Acknowledgements

Cryo-EM imaging was conducted at the Molecular Electron Microscopy Core facility at the University of Virginia, which is supported by the School of Medicine and built with the National Institutes of Health (NIH) grant G20-RR31199. In addition, the Titan Krios (SIG S10-RR025067) and K3/GIF (U24-GM116790) were purchased with the aid of the designated NIH grants. We thank Anchelique Mets and Ellen Hopmans (NIOZ) for technical support and mass spectral interpretations, respectively. Special thanks to Clay Fuqua for providing the *Agrobacterium tumefaciens* strain from which we obtained the T-pili. This work was supported by NIH Grant GM122510 (E.H.E.), GM138756 (F.W.) Welcome Trust Grant 215164/Z/18/Z (T.R.D.C), and l'Agence Nationale de la Recherche grant ANR-21-CE11-0001-01 (M.K.).

## Author contributions

L.B., V.C.-K., V.C., F.W., M.A.B.K., E.H.E., and M.K. designed the studies and experiments. V.C.-K. prepared the archaeal samples. J.B.P. and T.R.D.C. prepared the pED208 pili. J.M., V.C., and M.A.B.K. prepared the *A. tumefaciens* T-pili. S.S. analyzed the archaeal lipids, and I.L. analyzed the bacterial lipids. M.K. performed sequence analyses. L.B., E.H.E., and M.K. wrote the paper, with assistance from all authors.

## Competing interests

The authors declare no competing interests.
