## [Peer Review File · Nature Communications]

Archaeal DNA import apparatus is homologous to bacterial conjugation machineryReviewer #1 (Remarks to the Author):

This work presents 3 cryoEM structures of microbial surface filaments – 2 from archaea and 1 from a bacterium. The 2 archaeal filaments show structural homology to the bacterial filament, which has been shown previously to be a mating pilus. The work is interesting and will allow new avenues of inquiry.

A concern that I have (that can be remedied easily) is that it is possible to determine the structures of helical filaments incorrectly when the parameters are only slightly off. Therefore, the authors should supply supporting data showing how the helical parameters were determined:

- 1) Power spectra with Fourier Bessel orders highlighted
- 2) unbiased helical refinements (with no helical parameters applied) to show that the helical symmetry is the same as in the symmetrised maps
- 3) Power spectra (ideally in x/y and x/z) and FSCs for the unbiased and symmetrised refinements
- 4) a heatmap for the helical symmetry search for each filament

Also, do the authors know that the 2 archaeal filaments are definitely conjugative pili that mediate DNA transfer? They don't show evidence for DNA transfer in the work, and I didn't notice an obvious reference to it for these particular systems. Whilst it could probably be assumed that the filaments transport DNA due to the structural homology to other systems (where transfer has been demonstrated) – the authors should either make it clear that it has been demonstrated experimentally for these specific systems (if it has) or express a little more caution. I find this particularly important for the Pc pilus, which has a very narrow diameter lumen, which the authors discuss starting at L307. They state that there is no experimental evidence for transport of dsDNA (L311), but is there any evidence for DNA transport at all? Examples include the title, Line 103, L114, discussion e.g. L360, L377.

In addition, these are the specific points that I would recommend addressing:

1. Line 97 – “the ” is missing between “whether” and “Ced”
2. It wasn't clear to me why *A. tumefaciens* is a model system
3. L106 – “the subunit” is vague – pilin subunit?
4. L111 – the proposal that the system has been domesticated to allow survival in extreme environments is interesting, but it is a hypothesis and it is written as fact.
5. L120 – refers to Fig. 1A, but this doesn't follow the text as the panel doesn't reveal that this is a new type of filament not previously seen in archaea.
6. L120 – should be “filament”, not plural.
7. L123 – refers to Fig. 1A-C, but this doesn't follow the text either – it doesn't demonstrate that the protein is an ortholog of CedA1.
8. L124 – reference needed.
9. L130-131 and Fig. 1 – it is stated that the pili are structurally similar. At first glance, panels B and E don't look very similar, although the individual subunits do in panel C. Suggest writing this more clearly, plus comparing the filaments in more detail at this point, highlighting similarities and differences and stating the rise and twist values in the main text, as per the bacterial filament.
10. L133 “a” is missing between “display” and “similar”.
11. L133. Description of the Ted subunit nomenclature needs to come in earlier because proteins are named as Ted the figures, but as Pcal_0765 (or similar) in the text referring to them. Also L144, L147.
12. L137 – are the data for SP cleavage prediction shown? No figure panel referred to here.
13. L140 – how were the homologs identified? Sequence or structural homologs?
14. L147 – “to” missing between “related” and “Vir”
15. L 146 – Hard to follow how the profile-profile comparisons against the PBD database were done. Suggest more detail on HH-suite in legend or methods.
16. L150 – which part of Fig. 1G is being referred to? Hard to follow. Which ones are CedA-like and CedA1-like? Doesn't help that there's no legend to panel G.
17. L154 – is there a homologue of CedA2?
18. L191 – no comma needed.
19. L201 and Fig. 2F – “both non-protein densities could be fit with this lipid” – but the two look clearly different. Is this the same lipid in different folds? Needs to be explained more clearly.

20. L204 – can't see the angled appearance clearly in panel G.
21. L205 – can't see interaction with partially folded lipid in panel E. Make panels A and E bigger?
22. L232 – no need for the word "entirely".
23. L236 – filament is only referred to as VirB2 T-pili here. Consistency with nomenclature needed.
24. L253 – refer to a figure to show overall quite positive.
25. L258 – "The T-pilus is known..." sentence construction not right.
26. L259 – the transport of ssDNA-VirE2 needs a comment. Looking at the size of VirE2 (AlphaFold prediction) and the size of the lumen of the filament, it is hard to see how VirE2 could fit through.
27. L262 – comment on how a more electropositive lumen would allow transport of "all sorts of substrates" needed. Does this mean a more diverse set of substrates? Why should this be? And how is this reconciled with the transport of DNA, which is negative, and might therefore stick to the positive lumen?
28. L266 – comment on whether the two residues that are supposed to be joined in the cyclic form are in close proximity here.
29. L276 – refer to figure.
30. L289 – how was global structural similarity performed?
31. L302 – 303 – I wouldn't say that structures that possess lumen sizes ranging from 16Å to 26Å are comparable; they are quite different.
32. L511-514 reads oddly, like a set of instructions.
33. L528 – section heading wording not right.

Figures and legends

34. Fig. 1 – better to describe *A. pernix* first in the legend as it appears first in the figure. No legend for G.
35. Fig. 2 – why not make lipid colours consistent between panels? For some reason they change colour in panels C and G. In E, are these the same lipid in different forms, or different lipids? Same as point 20.
36. Fig. 3 – not a good choice of colours for colour blind readers. Worse in S1B. Why isn't the lipid modelled in here as per the other filaments?
37. L660 – "so we are looking at" is not very scientific.
38. I'm not convinced by using 2D classes to determine the absence or presence of glycosylation. It is also not clear where some of the data in Fig. S4 come from. Also, my understanding is that T4P are glycosylated – are these Ng and Pa versions not? Could the peripheral density be accounted for by something else (flexibility?) rather than glycosylation?
39. Fig. S5 – why are only 2 of the 3 filaments compared?
40. Fig. 5 doesn't add a lot as it stands and probably doesn't need to be in the main text. Can't see the DNA.
41. Table S2 – it has not been described how many pili were counted, or how many biological replicates performed.

Reviewer #2 (Remarks to the Author):

Conjugation pili in prokaryotes play a part in horizontal gene transfer which is responsible for the spread of antibiotic resistance among human pathogens. They are involved in establishing a mating junction between a donor and a recipient cell. While much has been studied in the conjugation machinery in bacteria, less is known about the conjugation in archaea. In this manuscript, the authors determined high-resolution cryo-EM structures of three conjugative pili, two from hyperthermophilic archaea, *A. pernix* and *P. calidifontis*, and one from the bacterium *A. tumefaciens* (T-pilus). The authors show that the archaeal pili are homologous to bacterial conjugation pili and reveal detailed pilin-lipid interactions. Moreover, the authors claim that the conjugation machinery in hyperthermophilic archaea has been domesticated and hypothesize that pili-mediated DNA exchange between hyperthermophilic archaea facilitates DNA repair by homologous recombination.

Although the structure of the T-pilus reported in this paper is very similar to the ones solved by

two other groups previously, the authors have determined the T-pilin-bound lipid to be dominantly PE, as compared to the other two papers, which claimed the dominant T-pilin-bound lipid is PC or PG, respectively. The structures of the archaeal conjugation pili reported in this paper are more interesting! The authors revealed very different pilin-lipid interactions in each asymmetric unit of the different conjugation pili. Furthermore, all the conjugation pili that have the structures solved so far have a narrow internal channel with an inner diameter that can only fit through an ssDNA, particularly for the *P. calidifontis* pili, which has a lumen significantly narrower than the other conjugation pili. All these pili structures are beautifully presented in this manuscript and give a more complete picture of the conjugation in prokaryotes. I am in favor of its publication in Nature Communications with some minor clarification needed from the authors. Please see below:

Line 124, Please provide a reference to the sentence "previously thought to be an integral membrane protein".

Line 262, The authors state that "It is therefore tempting to suggest that the more electropositive luminal environment is favorable for all sorts of substrates which pass through the lumen." It is not clear to me how a positively-charged lumen favors substrate passing through. Can the authors elaborate on this?

In addition, several conjugative pili have a lumen that is negatively charged, which the authors hypothesize that it would provide a "repulsive force" to keep the DNA away from the lumen wall. However, the lumen of the T-pilus is positively charged. Isn't this contradictory to the authors' hypothesis for the easy passage?

Response to the Reviewers

Reviewer #1 (Remarks to the Author):

This work presents 3 cryoEM structures of microbial surface filaments – 2 from archaea and 1 from a bacterium. The 2 archaeal filaments show structural homology to the bacterial filament, which has been shown previously to be a mating pilus. The work is interesting and will allow new avenues of inquiry.

A concern that I have (that can be remedied easily) is that it is possible to determine the structures of helical filaments incorrectly when the parameters are only slightly off. Therefore, the authors should supply supporting data showing how the helical parameters were determined:

- 1) Power spectra with Fourier Bessel orders highlighted
- 2) unbiased helical refinements (with no helical parameters applied) to show that the helical symmetry is the same as in the symmetrised maps
- 3) Power spectra (ideally in x/y and x/z) and FSCs for the unbiased and symmetrised refinements
- 4) a heatmap for the helical symmetry search for each filament

RESPONSE: We agree that more data should have been shown to justify the helical symmetries that were used. We have now added Supplementary Fig. 10 which shows averaged power spectra for each of the three filaments reconstructed, with the indexing (Bessel orders) labeled. We disagree with the utility of points 2-4 above, and have written in a number of previous papers that the problem is worse than what the reviewer suggests (“that it is possible to determine the structures of helical filaments incorrectly when the parameters are only slightly off”), and that it is actually possible to determine the structures of helical filaments incorrectly when the parameters are substantially off. For example, this is shown in a paper entitled “Ambiguities in helical reconstruction” (Egelman, 2014) as well as in a paper entitled “Cryo-EM is a powerful tool, but helical applications can have pitfalls” (Egelman and Wang, 2021). Points 2-4 above stem from the notion that one can generate an asymmetric reconstruction, with no helical symmetry imposed, and then find the correct helical symmetry within that asymmetric volume. This approach is actually available and documented in cryoSPARC. The lack of novelty did not stop another group from publishing the same approach in a recent paper, “Helical indexing in real space” (Sun et al., 2022).

The problem with this approach is that the correct or incorrect helical symmetry has already been locked in when one generates the asymmetric (or “unbiased”) reconstruction, and the asymmetric reconstruction is not unique. That is, just as one can take an ensemble of images and generate different 2D averages from this same ensemble (including one that will look like Albert Einstein if Einstein is used as a reference) (Henderson, 2013), one can generate different 3D volumes from this ensemble with the assignment of different Euler angles and translational parameters to each image. If one uses a reference volume that has a particular helical symmetry, then this symmetry will “appear” in the asymmetric reconstruction. If one uses a featureless initial reference, assignments of Euler angles to each segment will be stochastic but one can converge on different symmetries in different reconstructed volumes

using the same image stack. Thus, we were very impressed when cryoSPARC determined the correct helical symmetry from a data set of segments using this approach. But rerunning everything with the same parameters, the correct “solution” was not even among the top 20 found. This is obviously because the correct symmetry had been locked into the first asymmetric reconstruction, while an incorrect symmetry had been locked into the second volume. Once the incorrect symmetry gets locked in, there is no algorithm or search procedure that can find the correct symmetry in this volume, and no heatmap will show the correct minimum. Unless there is some “magic” that can be employed to assign Euler angles to helical segments correctly, all such *ab initio* unbiased asymmetric volumes will have the potential to have an incorrect symmetry. The reason that there is no simple solution to this problem is not a failure of current methods, it is simply a mathematical ambiguity such that there are multiple solutions that are indistinguishable (in terms of residuals, etc.) at some finite resolution.

To illustrate this further, we submitted six different “unbiased” reconstruction volumes from three different specimens to the server described in Sun et al., which is essentially the same as a heatmap generated by cryoSPARC. Due to the stochastic elements in this reconstruction process (discussed above), the two volumes from each specimen are never identical. Here are the dismal results for the best symmetry found in each of the two unbiased volumes:

Filament A: true symmetry has 38.46° twist, 5.75 Å rise, C1 (pitch=53.8 Å)

Volume 1: -35.95°, 4.14 Å, C1 (pitch = 41.5 Å)

Volume 2: -41.37°, 4.75 Å, C1 (pitch = 41.3 Å)

Filament B: true symmetry has -83.41° twist, 4.11 Å rise, C2 (pitch of 2-start=17.7 Å)

Volume 1: -9.51°, 8.17 Å, C1, (pitch=309.3 Å)

Volume 2: 13.67°, 8.21 Å, C1, (pitch=216.2 Å)

Filament C: true symmetry has 22.59° twist, 0.305 Å rise, C1 (pitch=4.86 Å)

Volume 1: 5.47°, 19.61 Å, C1 (pitch=1,290.6 Å)

Volume 2: 5.93 Å, 19.57 Å, C1 (pitch=1,188.1 Å)

We also do not think that there is utility in comparing the power spectra from the actual segments (1) with power spectra from the reconstruction (3) as this reinforces the notion that this is a validation of the reconstruction, when it is not. It has been shown (Egelman, 2010) that one can generate power spectra from multiple wrong helical symmetries that are indistinguishable from the power spectrum with the correct helical symmetry. Ultimately, the validation that we have correctly determined the helical symmetry comes from the map and the model.

Also, do the authors know that the 2 archaeal filaments are definitely conjugative pili that mediate DNA transfer? They don't show evidence for DNA transfer in the work, and I didn't notice an obvious reference to it for these particular systems. Whilst it could probably be assumed that the filaments transport DNA due to the structural homology to other systems (where transfer has been demonstrated) – the authors should either make it clear that it has been demonstrated experimentally for these specific systems (if it has) or express a little more

caution. I find this particularly important for the Pc pilus, which has a very narrow diameter lumen, which the authors discuss starting at L307. They state that there is no experimental evidence for transport of dsDNA (L311), but is there any evidence for DNA transport at all? Examples include the title, Line 103, L114, discussion e.g. L360, L377.

RESPONSE: There is unambiguous experimental evidence that the Ced system is involved in the intraspecies DNA transfer, specifically, import of chromosomal DNA (van Wolferen et al., 2016). In the revised manuscript, we made this more explicit. In the case of the Ted system, our conclusion is based on the structural similarity between the Ted and Ced as well as bacterial conjugative pili, which are known to transfer DNA substrates. Nevertheless, experimental evidence for the Ted system is lacking and in the absence of the genetic tools for any member of the order Thermoproteales, direct experiments are not possible. Thus, we toned down most of the claims, as follows:

line 103:

“We present high-resolution structures of three conjugative pili...”

Changed to:

“We present high-resolution structures of two putative conjugative pili from hyperthermophilic archaea, *A. pernix* and *Pyrobaculum calidifontis*, and a bacterial conjugation pilus ...”

line 114:

“Identification of DNA transfer pili in hyperthermophilic archaea...”

Changed to:

“Identification of putative DNA transfer pili in hyperthermophilic archaea...”

Line 359:

“...a unique feature of bacterial and archaeal conjugation pili”

Changed to:

“...a unique feature of bacterial conjugation pili and their archaeal homologs”

line 377:

“Nevertheless, our data indicates that archaeal pili, similar to their bacterial counterparts, must transfer ssDNA.”

Changed to:

“Nevertheless, our data indicate that archaeal pili, similar to their bacterial counterparts, most likely transfer ssDNA.”

In addition, these are the specific points that I would recommend addressing:

1. Line 97 – “the “ is missing between “whether” and “Ced”

RESPONSE: Fixed

2. It wasn't clear to me why *A. tumefaciens* is a model system

RESPONSE: We agree, and have put in a number of references in support of this statement.

3. L106 – “the subunit” is vague – pilin subunit?

RESPONSE: We have changed this to “the pilin subunit”

4. L111 – the proposal that the system has been domesticated to allow survival in extreme environments is interesting, but it is a hypothesis and it is written as fact.

RESPONSE: We have clarified that this is a hypothesis.

5. L120 – refers to Fig. 1A, but this doesn’t follow the text as the panel doesn’t reveal that this is a new type of filament not previously seen in archaea.

RESPONSE: We have changed the text to read: “we identified a new type of filament (Fig. 1A), not previously observed in archaea”

6. L120 – should be “filament”, not plural.

RESPONSE: Fixed

7. L123 – refers to Fig. 1A-C, but this doesn’t follow the text either – it doesn’t demonstrate that the protein is an ortholog of CedA1.

RESPONSE: We have removed the reference to Fig. 1A-C.

8. L124 – reference needed.

RESPONSE: We have inserted the reference to van Wolferen *et al.*

9. L130-131 and Fig. 1 – it is stated that the pili are structurally similar. At first glance, panels B and E don’t look very similar, although the individual subunits do in panel C. Suggest writing this more clearly, plus comparing the filaments in more detail at this point, highlighting similarities and differences and stating the rise and twist values in the main text, as per the bacterial filament.

RESPONSE: We have clarified this comparison as suggested.

10. L133 “a” is missing between “display” and “similar”.

RESPONSE: Fixed

11. L133. Description of the Ted subunit nomenclature needs to come in earlier because proteins are named as Ted the figures, but as Pcal_0765 (or similar) in the text referring to them. Also L144, L147.

RESPONSE: We have introduced TedC nomenclature earlier in the text, as suggested.

12. L137 – are the data for SP cleavage prediction shown? No figure panel referred to here.

RESPONSE: In the revised manuscript, we included a new panel in Supplementary figure 1 (panel b) with the results of the SignalP prediction for the signal peptide and signal peptidase cleavage site and also provide the probability of the cleavage site prediction in the main text.

13. L140 – how were the homologs identified? Sequence or structural homologs?

RESPONSE: They were identified by sequence similarity, and this has been clarified in the text.

14. L147 – “to” missing between “related” and “Vir”

RESPONSE: Fixed

15. L 146 – Hard to follow how the profile-profile comparisons against the PBD database were done. Suggest more detail on HH-suite in legend or methods.

RESPONSE: We have clarified this in Methods.

16. L150 – which part of Fig. 1G is being referred to? Hard to follow. Which ones are CedA-like and CedA1-like? Doesn't help that there's no legend to panel G.

RESPONSE: We apologize for not including the legend to panel G. It has now been added, and this legend clarifies which are CedA-like (green) and CedA1-like (red).

17. L154 – is there a homologue of CedA2?

RESPONSE: CedA2 is not conserved even among Ced systems from different species and homologs or even counterparts of this protein are not identifiable in the Ted system. This is now mentioned in the manuscript text.

18. L191 – no comma needed.

RESPONSE: Fixed

19. L201 and Fig. 2F – “both non-protein densities could be fit with this lipid” – but the two look clearly different. Is this the same lipid in different folds? Needs to be explained more clearly.

RESPONSE: We have clarified that the densities are consistent with the same lipid in two different conformations.

20. L204 – can't see the angled appearance clearly in panel G.

RESPONSE: We have clarified this.

21. L205 – can't see interaction with partially folded lipid in panel E. Make panels A and E bigger?

RESPONSE: We agree, and have modified the figure.

22. L232 – no need for the word “entirely”.

RESPONSE: Removed.

23. L236 – filament is only referred to as VirB2 T-pili here. Consistency with nomenclature needed.

RESPONSE: Fixed, and we now consistently refer to it as the T-pilus.

24. L253 – refer to a figure to show overall quite positive.

RESPONSE: We have added a reference to Supp. Fig. 6.

25. L258 – “The T-pilus is known...” sentence construction not right.

RESPONSE: We have fixed this sentence.

26. L259 – the transport of ssDNA-VirE2 needs a comment. Looking at the size of VirE2 (AlphaFold prediction) and the size of the lumen of the filament, it is hard to see how VirE2 could fit through.

RESPONSE: We have added a sentence suggesting that VirE2 needs to be partially unfolded to be accommodated in the lumen.

27. L262 – comment on how a more electropositive lumen would allow transport of “all sorts of substrates” needed. Does this mean a more diverse set of substrates? Why should this be? And how is this reconciled with the transport of DNA, which is negative, and might therefore stick to the positive lumen?

RESPONSE: We have clarified this and introduced the speculation that while the lumen of other bacterial mating pili may have evolved to be optimal for DNA transport, the lumen of the T-pilus, also used for the transport for a diverse set of other substrates, has a positive electrostatic potential that would still allow DNA transfer, but not be optimal due to the greater friction resulting from DNA sticking to the walls. Obviously, this would suggest that the other

substrates are likely to have overall positive electrostatic surfaces, and it has been noted that these effector proteins have positively charged C-terminal domains (Li and Christie, 2018). We have added a reference to a review article discussing the other substrates transported through the T-pilus.

28. L266 – comment on whether the two residues that are supposed to be joined in the cyclic form are in close proximity here.

RESPONSE: We have clarified the text. The two residues suggested to be covalently linked in Eisenbrandt et al. (1999) were Gln1 and Gly74 (using our numbering for the mature pilin). Neither of these residues are modeled in our structure due to lack of any density for them, presumably due to disorder. We have explained, however, that the first and last residues modeled, Gly5 and Gly73, are too far apart (~ 34 Å) to preclude the possibility that the five missing residues could be forming a cyclic subunit.

29. L276 – refer to figure.

RESPONSE: We have added references to Supp. Fig. 7 and Supp. Table S2

30. L289 – how was global structural similarity performed?

RESPONSE: We have clarified that this was done using Dali Z-scores.

31. L302 – 303 – I wouldn't say that structures that possess lumen sizes ranging from 16A to 26A are comparable; they are quite different.

RESPONSE: We agree, and have changed this text.

32. L511-514 reads oddly, like a set of instructions.

RESPONSE: We agree, and have changed this text.

33. L528 – section heading wording not right.

RESPONSE: We have changed this to “Mass spectrometry and lipidomics of *A. pernix*”

Figures and legends

34. Fig. 1 – better to describe *A. pernix* first in the legend as it appears first in the figure. No legend for G.

RESPONSE: We apologize for the oversight. The legend descriptions of *A. pernix* and *P. calidifontis* are now swapped and the missing legend for panel G is now added.

35. Fig. 2 – why not make lipid colours consistent between panels? For some reason they

change colour in panels C and G. In E, are these the same lipid in different forms, or different lipids? Same as point 20.

RESPONSE: Thank you for this suggestion. The lipid colors were fixed to maintain consistency.

36. Fig. 3 – not a good choice of colours for colour blind readers. Worse in S1B. Why isn't the lipid modelled in here as per the other filaments?

RESPONSE: In Fig. 3 the red and green colors were adjusted to a colorblind palette. Figure S1B colors were also changed. We have added the lipid model to Fig. 3D.

37. L660 – “so we are looking at” is not very scientific.

RESPONSE: We have changed this to “and we are looking at the lumen”.

38. I'm not convinced by using 2D classes to determine the absence or presence of glycosylation. It is also not clear where some of the data in Fig. S4 come from. Also, my understanding is that T4P are glycosylated – are these Ng and Pa versions not? Could the peripheral density be accounted for by something else (flexibility?) rather than glycosylation?

RESPONSE: First, we are not using 2D class averages in Supp. Fig. 4. All of the data shown come from projections of the 3D reconstructions using a high threshold (left) or a low threshold (right). Yes, a few residues in the bacterial T4P are glycosylated, but this would not generate the large cloud of surrounding density seen in an extensively glycosylated filament, LAL14/1 (Wang et al., 2019). For the peripheral density to be due to flexibility of the protein, one would need to have a large number of residues not present in the model, and expected to be on the surface of the filament, that could be generating this density. But such disordered residues do not exist in any of the filaments shown here.

39. Fig. S5 – why are only 2 of the 3 filaments compared?

RESPONSE: Actually, in Supplementary Fig. 6 we are only comparing 1 of the 3 filaments in the paper, the T-pilus, with a previously published model of the pED208 conjugation pilus (Costa et al., 2016). We have not shown any electrostatic potential surface for the two archaeal filaments as this surface would be quite sensitive to the lipid headgroup, as previously shown (Costa et al., 2016). Since we do not know the composition of the lipid headgroups in the archaeal filaments, they are not being compared here.

40. Fig. 5 doesn't add a lot as it stands and probably doesn't need to be in the main text. Can't see the DNA.

RESPONSE: We agree, and have moved this figure to Supplementary Fig. 9. We have also made the DNA more visible.

41. Table S2 – it has not been described how many pili were counted, or how many biological replicates performed.

RESPONSE: Each condition/treatment was replicated three times for both the T-pilus and pED208. Although the pili were not counted, over 15 grid squares were chosen at random and examined for each sample to ensure a broad representation of pilus morphology across the entire electron microscopy grid. This has now been noted in the paper.

Reviewer #2 (Remarks to the Author):

Conjugation pili in prokaryotes play a part in horizontal gene transfer which is responsible for the spread of antibiotic resistance among human pathogens. They are involved in establishing a mating junction between a donor and a recipient cell. While much has been studied in the conjugation machinery in bacteria, less is known about the conjugation in archaea. In this manuscript, the authors determined high-resolution cryo-EM structures of three conjugative pili, two from hyperthermophilic archaea, *A. pernix* and *P. calidifontis*, and one from the bacterium *A. tumefaciens* (T-pilus). The authors show that the archaeal pili are homologous to bacterial conjugation pili and reveal detailed pilin-lipid interactions. Moreover, the authors claim that the conjugation machinery in hyperthermophilic archaea has been domesticated and hypothesize that pili-mediated DNA exchange between hyperthermophilic archaea facilitates DNA repair by homologous recombination.

Although the structure of the T-pilus reported in this paper is very similar to the ones solved by two other groups previously, the authors have determined the T-pilin-bound lipid to be dominantly PE, as compared to the other two papers, which claimed the dominant T-pilin-bound lipid is PC or PG, respectively. The structures of the archaeal conjugation pili reported in this paper are more interesting! The authors revealed very different pilin-lipid interactions in each asymmetric unit of the different conjugation pili. Furthermore, all the conjugation pili that have the structures solved so far have a narrow internal channel with an inner diameter that can only fit through an ssDNA, particularly for the *P. calidifontis* pili, which has a lumen significantly narrower than the other conjugation pili. All these pili structures are beautifully presented in this manuscript and give a more complete picture of the conjugation in prokaryotes. I am in favor of its publication in Nature Communications with some minor clarification needed from the authors. Please see below:

Line 124, Please provide a reference to the sentence “previously thought to be an integral membrane protein”.

RESPONSE: We have inserted the reference to van Wolferen *et al.*

Line 262, The authors state that “It is therefore tempting to suggest that the more electropositive luminal environment is favorable for all sorts of substrates which pass through

the lumen.” It is not clear to me how a positively-charged lumen favors substrate passing through. Can the authors elaborate on this?

In addition, several conjugative pili have a lumen that is negatively charged, which the authors hypothesize that it would provide a “repulsive force” to keep the DNA away from the lumen wall. However, the lumen of the T-pilus is positively charged. Isn’t this contradictory to the authors’ hypothesis for the easy passage?

RESPONSE: To address this questions we have introduced the speculation that while the lumen of other bacterial mating pili may have evolved to be optimal for DNA transport, the lumen of the T-pilus, also used for the transport for a diverse set of other substrates, has a positive electrostatic potential that would still allow DNA transfer, but not be optimal due to the greater friction resulting from DNA sticking to the walls. Obviously, this would suggest that the other substrates are likely to have overall positive electrostatic surfaces, and it has been noted that these effector proteins have positively charged C-terminal domains (Li and Christie, 2018). We have added a reference to this review article discussing the other substrates transported through the T-pilus.

References

Costa, T.R., Ilangovan, A., Ukleja, M., Redzej, A., Santini, J.M., Smith, T.K., Egelman, E.H., and Waksman, G. (2016). Structure of the Bacterial Sex F Pilus Reveals an Assembly of a Stoichiometric Protein-Phospholipid Complex. *Cell* *166*, 1436-1444 e1410.

Egelman, E.H. (2010). Reconstruction of helical filaments and tubes. *Meth. Enzymol.* *482*, 167-183.

Egelman, E.H. (2014). Ambiguities in Helical Reconstruction. *eLife* *3*:e04969
doi:10.7554/eLife.04969

Egelman, E.H., and Wang, F. (2021). Cryo-EM is a powerful tool, but helical applications can have pitfalls. *Soft Matter* *17*, 3291-3293.

Henderson, R. (2013). Avoiding the pitfalls of single particle cryo-electron microscopy: Einstein from noise. *Proc. Natl. Acad. Sci. U.S.A.* *110*, 18037-18041.

Li, Y.G., and Christie, P.J. (2018). The Agrobacterium VirB/VirD4 T4SS: Mechanism and Architecture Defined Through In Vivo Mutagenesis and Chimeric Systems. *Curr Top Microbiol Immunol* *418*, 233-260.

Sun, C., Gonzalez, B., and Jiang, W. (2022). Helical Indexing in Real Space. *Sci Rep* *12*, 8162.

van Wolferen, M., Wagner, A., van der Does, C., and Albers, S.V. (2016). The archaeal Ced system imports DNA. *Proc. Natl. Acad. Sci. U.S.A.* *113*, 2496-2501.

Wang, F., Cvirkaite-Krupovic, V., Kreutzberger, M.A.B., Su, Z., de Oliveira, G.A.P., Osinski, T., Sherman, N., DiMaio, F., Wall, J.S., Prangishvili, D., Krupovic, M., and Egelman, E.H. (2019). An extensively glycosylated archaeal pilus survives extreme conditions. *Nat Microbiol* *4*, 1401-1410.

Reviewer #1 (Remarks to the Author):

I am satisfied with the response to my comments.

Just one point (number 38) was potentially missed. A reference to where the Ng and Pa data used in the comparison is missing.

Response to the reviewer

I am satisfied with the response to my comments.

Just one point (number 38) was potentially missed. A reference to where the Ng and Pa data used in the comparison is missing.

We have modified the legend to Supp. Fig. 4 to now provide this information, as well as information for the LAL14/1 pilus:

Supplementary Fig. 4. Projections of the 3D reconstructions at different thresholds reveal if pili are extensively glycosylated. T4P pili of *Neisseria gonorrhoeae* (**A**, EMDB deposition EMD-8739, [<https://www.emdataresource.org/EMD-8739>]) and *Pseudomonas aeruginosa* (**B**, EMDB deposition EMD-8740, [<https://www.emdataresource.org/EMD-8740>]) are shown as negative controls, whereas the highly glycosylated type IV pilus of *Saccharolobus islandicus* LAL14/1 (**C**, EMDB deposition EMD-0397, [<https://www.emdataresource.org/EMD-0397>]) shows peripheral density at low threshold, and serves as a positive control. Both the Ted pilus of *P. calidifontis* (**D**) and the Ced pilus of *A. pernix* (**E**) show some surrounding density, which could represent glycosylation. In contrast, the T-pilus of *A. tumefaciens* (**F**) shows no obvious surrounding density at low threshold and behaves like controls A and B.